# Collaborative and Efficient Personalization with Mixtures of Adaptors

Abdulla Jasem Almansoori    Samuel Horváth    Martin Takáč

Mohamed bin Zayed University of Artificial Intelligence, Abu Dhabi, UAE

`firstname.lastname@mbzuai.ac.ae`

Heterogenous data is prevalent in real-world federated learning. We propose a parameter-efficient framework, Federated Low-Rank Adaptive Learning (FLoRAL), that allows clients to personalize in groups by mixing between low-rank adaptors, where the mixtures are client-specific. FLoRAL is a model parameterization that casts personalized federated learning as a multi-task learning problem, with weight sharing as an implicit regularizer. It is memory-efficient, as the personalized parameters (i.e., base model + adaptors) are all federated. Our results show that FLoRAL can generalize better than a mixture of full models when data are scarce. It can also consistently personalize better than models with a locally tuned adaptor per client. This demonstrates the benefits of "federated personalization" and its robustness against overfitting. We derive the convergence rates and show theoretically that FLoRAL can lead to better variance reduction of the base model's gradients. [1]

## 1. Introduction

In Federated Learning (FL), clients serve as decentralized holders of private data, and they can collaborate via secure aggregation of model updates, but one of the main challenges is the heterogeneity of the clients [1]. For example, heterogeneity can be in terms of data distributions (statistical heterogeneity) or client capabilities (system heterogeneity) [2]. In this work, we are interested in a statistical heterogeneity where labels are predicted differently across clients. In particular, this can be viewed under the lens of multi-task learning [3] or clustering [4] such that there are only a few ground-truth tasks or clusters across all clients.

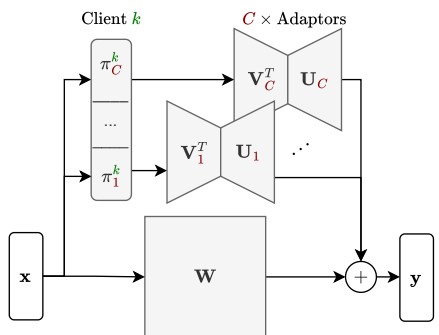

Figure 1: Personalization for client $k$ by mixing $C$ adaptors.

The central assumption in our work is that the personalized models across clients should be similar enough to benefit from collaboration, but they also need to be sufficiently different and expressive to fit and generalize on their personal data. The differences between clients can be thought of as 1) statistical in terms of data (e.g., shifts in distributions) or structural in terms of model (e.g., structured differences in subsets of parameters). To learn these differences *efficiently*, we often assume that they are *low-complexity* differences.

Most approaches maintain that the personalized models are either close in distance to the global model via proximal regularization [5–8] or meta-learning [9], or that the personalized models belong to a cluster of models [3, 4]. Other approaches also assume model heterogeneity, where clients might have a local subset of parameters that are not averaged [10, 11] where it can personalize to the local task by construction [12]. For example, a specific subset of parameters can be chosen to be the last layer or some added *adaptors*.

---

[1]Code: `https://github.com/zeligism/FLoRAL`

Second Conference on Parsimony and Learning (CPAL 2025).

Fine-tuning works particularly well for personalization [13]. One well-known example of efficient fine-tuning is Low-Rank Adaptors (LoRA) [14], which are used to personalize large language models on different tasks. The fine-tuning is done on an additive low-rank matrices instead of the full matrices. Thus, the personalized models differ from the base model only in low-rank matrices. Inspired by the efficiency of low-rank adaptors in multi-task learning for language models and the idea that fine-tuning changes parameters along a low-dimensional intrinsic subspace [15, 16], we use low-rank adaptors in the FL setting and show that they can offer significant improvements with a relatively small memory budget.

Thus, instead of regularizing the complexity of a personalized model by its proximity to a reference solution or clustering full models, we explicitly parameterize the personalized models as having a few common low-rank differences from the global model. This is done by introducing a small number of low-rank adaptors per layer and a mixture vector per client that mixes between those adaptors. Thus, it implicitly regularizes the personal models through weight-sharing with low-rank differences. Our approach explicitly constrains the complexity of the difference between the global model and the personalized model, and casts the problem of learning these differences as a multi-task learning problem. Its main benefit is that the low-rank adaptors can also be federated and collaboratively learned. The number of local personalization parameters per client (i.e., the mixture vectors) is minimal, which means our approach can be efficiently employed in the cross-device setting.

**Contributions** Here, we summarize our contributions:

1. We propose the *Federated Low-Rank Adaptive Learning (FLoRAL)*, an efficient and lightweight FL framework for personalization. It acts as an extension to multi-task learning algorithms that are specifically designed for FL.

2. Perhaps counter-intuitively, we show experimentally that a model with a mixture of adaptors can beat a mixture of models, even though the number of parameters is significantly larger, e.g., 9x larger. Also, a model with a mixture of adaptors on stateless clients (e.g., see Section 5) can generalize better than a model with a dedicated fine-tuned adaptor on stateful clients. This is a perfect demonstration of the efficiency of FLoRAL and the benefits of collaborative learning.

3. We release the code for this framework, which includes plug-and-play wrappers for PyTorch models [17] that are as simple as `Floral(model, rank=8, num_clusters=4)`. We also provide minimal extensions of Flower client and server modules [18], making the adoption of our method in practice and reproducing the experiments seamless and easy.

4. We run various experiments and ablation studies showing that our FLoRAL framework is efficient given resource constraints in terms of relative parameter increase.

5. We provide the convergence rate for local SGD on a multi-task objective with learnable router and highlight the difficulties that arise from aggregation mismatch. We also provide an extended analysis in the appendix showing better variance reduction from weight sharing.

## 2. Related Work

**Multi-task Learning** Our problem can be seen as a multi-task learning problem in which the solutions share a base model. The closest work to ours in this respect is FedEM [3], which works by assigning to each client a personalized mixture vector that mixes between a small number of full models such that each model solves one task. FedEM then proceeds with an algorithm based on expectation maximization. One problem is that their approach assumes that the full models should be mixed. In contrast, we assume that the mixed components are only the adaptors, which constitute a small fraction of the model and are thus much more efficient in terms of memory. Other related works on clustering include IFCA [19], FedSoft [20], and Federated-Clustering [4]. The main difference from our work is that we only cluster a small component of the whole model, allowing the clients to benefit from having a shared base model that is learned among all the clients.

**Personalization** Another approach to personalization is by introducing a proximal regularizer with respect to a reference model. Ditto [5] is a stateful algorithm that trains the local models by

solving a proximal objective with respect to a reference model. The reference model is the FedAvg solution, which is attained concurrently by solving the non-regularized objective. Meta-learning approaches, inspired by Finn et al. [21], can extend naturally to personalization. For example, Fallah et al. [9] propose to solve a local objective that is an approximate solution after one local gradient step. Meta-learning also assumes that the local solutions are close to the FedAvg solution as they mimic fine-tuning from the FedAvg solution in some sense. In our approach, we do not assume that the clients are stateful nor that the FedAvg solution is meaningful or close to any of the local solutions. We assume that the local models can benefit from collaboration but still allow for personalization via different mixtures, which is much more memory efficient and can be managed by the server.

**LoRA** Using mixture of LoRAs in FL is not new due to their popularity. The idea of mixing LoRAs has been explored recently [22] for language models. SLoRA [23] focuses on parameter-efficient fine-tuning after federated training and thus does not federate the adaptors. Both FedLoRA [24] and pFedLoRA [25] assume that the LoRAs are not federated as well, where they both also introduce a specific two-stage algorithm to train those LoRAs. The federated mixture of experts [26] trains an ensemble of specialized models, but they specialize in input rather than prediction. FedJETs [27] uses whole models as experts in addition to a pre-trained feature aggregator as a common expert that helps the client choose the right expert. Other works explore mixture of LoRAs [28, 29] for adaptation but in a different, non-collaborative context.

**Representation Learning** Other successful approaches in FL work by feature, prototype, or representation aggregation [30–32], which makes them orthogonal to our work as they work in the feature space.

## 3. Preliminary

**Notation** We denote $[N] = \{1, 2, \ldots, N\}$. We reserve some indices for specific objects: $k \in [K]$ is a superset index[2] denoting the client with $K$ being the number of clients, and $c \in [C]$ is a subset index denoting the cluster with $C$ being the number of clusters. The number of clients in cluster c is $K_c$. The client sampling distribution is $\mathcal{K}$, or $\mathcal{K}_c$ when given cluster $c$. The number of samples in client $k$ is $N^k$, and the total number of samples is $N = \sum_{k=1}^{K} N^k$. We will use bold lowercase characters to denote vectors, e.g., $\mathbf{w}$, and uppercase bold characters for matrices, e.g., $\mathbf{W}$. Further, $\mathbf{1}\{A\}$ is the indicator function of event $A$, and vec$(\cdot)$ is the vectorization operator. A simplex $\Delta^{C-1}$ is a space such that, for all $\boldsymbol{\pi} \in \Delta^{C-1}$, we have $\sum_{c=1}^{C} \boldsymbol{\pi}_c = 1$ and $\boldsymbol{\pi}_c \geq 0, \forall c \in [C]$.

### 3.1. Federated Learning

Federated learning (FL) is a framework for training a model on distributed data sources while the data remains private and on-premise. Let $K$ be the number of clients and the local loss function for client $k$ be $f^k(\mathbf{w})$. The global objective is

$$\min_{\mathbf{w}} \quad \mathbb{E}_{k \sim \mathcal{K}}[f^k(\mathbf{w})], \tag{FL}$$

where $\mathcal{K}$ is a client distribution with support $[K]$. The functions $f^k(\mathbf{w})$ can be stochastic as well. The most straightforward algorithm for optimizing (FL) is FedAvg [33], which proceeds in a cycle as follows: 1) send copies of the global model to the participating clients, 2) train the copies locally on the client's data, and then 3) send back the copies and aggregate them to get the new global model.

The objective (FL) assumes that a single global model can obtain an optimal solution that works for all the objectives, which is often not feasible due to heterogeneities in data distribution and system capabilities [1]. A natural approach would be to consider *personalized* solutions $\mathbf{w}^k$ for each client $k$, an approach called PersonalizedFL (PFL).

$$\min_{\{\mathbf{w}^k\}_{k=1}^{K}} \quad \mathbb{E}_{k \sim \mathcal{K}}[f^k(\mathbf{w}^k)] + \Gamma(\mathbf{w}^1, \cdots, \mathbf{w}^K). \tag{PFL}$$

---

[2]In general, we reserve the superset for clients and the subset for clusters.

Without the regularizer $\Gamma$, the objective would simply amount to local independent training for each client, so clients do not benefit from collaboration and can suffer from a low availability of data. Adding the regularizer $\Gamma$ helps introduce a collaboration incentive or inductive bias. For example, Ditto [5] adds a proximal regularizer $\Gamma(\mathbf{w}^1, \cdots, \mathbf{w}^K; \mathbf{w}^*) = \frac{\lambda}{2} \sum_{k=1}^{K} \|\mathbf{w}^k - \mathbf{w}^*\|^2$, where $\mathbf{w}^*$ is the solution of (FL). However, this assumes that a single global solution is a good enough center for *all* clients, which can be limiting for capturing real-world heterogeneities. An improvement on this assumption is to introduce more than one center, such that clients belonging to some group are close to its center. The problem of finding the group centers is called Clustered FL (CFL).

Let $C$ be the number of ground-truth clusters and assume that it is known. Let $\mathcal{K}_c$ be the client sampling distribution of cluster $c$. We can reformulate the objective to account for clusters as follows

$$\min_{\{\mathbf{w}_c\}_{c=1}^{C}} \quad \sum_{c=1}^{C} \mathbb{E}_{k \sim \mathcal{K}_c}[f^k(\mathbf{w}_c)]. \tag{CFL}$$

We can generalize the previous objectives under one objective by introducing (learnable) client mixtures $\boldsymbol{\pi}^k \in \Delta^{C-1}$ for all $k \in [K]$ with regularization $\Gamma$, e.g. for weight sharing, which we denote as Mixed Federated Learning (MFL)

$$\min_{\{\mathbf{w}_c\}_{c=1}^{C}, \{\boldsymbol{\pi}^k\}_{k=1}^{K}} \quad \sum_{c=1}^{C} \mathbb{E}_{k \sim \mathcal{K}} \left[\boldsymbol{\pi}_c^k f^k(\mathbf{w}_c)\right] + \Gamma(\{\mathbf{w}_c\}_{c=1}^{C}), \tag{MFL}$$

$$\text{s.t.} \quad \boldsymbol{\pi}^k \in \Delta^{C-1}, \forall k \in [K],.$$

We can see that local losses from different clusters are mixed differently according to each client. From this formulation, we can recover (CFL) by setting $\Gamma(\cdot) = 0$ and $\boldsymbol{\pi}_c^k = \mathbf{1}\{k \in \mathrm{supp}(\mathcal{K}_c)\}$, whereas (PFL) can be recovered by setting $C = K$ and $\boldsymbol{\pi}_c^k = \mathbf{1}\{k \in \mathrm{supp}(\mathcal{K}_c)\}$.

In our FLoRAL formulation, we use particular form of $\Gamma$, where we split $\mathbf{w}_c = [\mathbf{u}_c, \mathbf{a}_c]$ and define

$$\Gamma(\{\mathbf{w}_c\}_{c=1}^{C}) = \begin{cases} 0 & \text{if } \mathbf{u}_i = \mathbf{u}_j \; \forall i, j \in [C], \\ +\infty & \text{otherwise.} \end{cases}$$

This weight-sharing across clients is based on the inductive bias that the optimal personalized solutions have *low-complexity* differences across the population (i.e., differences that could be explained in a parameter-efficient way). Therefore, in the rest of the paper, we do not use $\Gamma(\{\mathbf{w}_c\}_{c=1}^{C})$, but we replace it with explicit parametrization, where $\mathbf{w}_c = (\mathbf{u}, \mathbf{a}_c)$. We refer to $\{\mathbf{a}_c\}_{c \in [C]}$ as adaptors. The final objective, which we call MFL with Weight Sharing (MFL-WS), is of the form

$$\min_{\mathbf{u}, \{\mathbf{a}_c\}_{c=1}^{C}, \{\boldsymbol{\pi}^k\}_{k=1}^{K}} \quad \sum_{c=1}^{C} \mathbb{E}_{k \sim \mathcal{K}} \left[\boldsymbol{\pi}_c^k f^k(\mathbf{u}, \mathbf{a}_c)\right] \tag{MFL-WS}$$

$$\text{s.t.} \quad \boldsymbol{\pi}^k \in \Delta^{C-1}, \forall k \in [K].$$

In the next section, we discuss the particular choice of adaptors.

## 3.2. Parameter-Efficient Adaptors

**Linear layer** Let $\mathbf{W} \in \mathbb{R}^{d_{out} \times d_{in}}$ be the base linear layer. The low-rank adaptor with rank $r$ is $\mathbf{L} := \mathbf{U}\mathbf{V}^\top$, where $\mathbf{U} \in \mathbb{R}^{m \times r}$ and $\mathbf{V} \in \mathbb{R}^{n \times r}$. We initialize $\mathbf{L}$ such that $\mathbf{U}$ is random (or initialized similarly to $\mathbf{W}$) and $\mathbf{V}$ is zero. The adaptive layer is then $\tilde{\mathbf{W}} := \mathbf{W} + \mathbf{L} = \mathbf{W} + \mathbf{U}\mathbf{V}^\top$

**Relative parameter budget** It is easy to see that the number of parameters in a linear LoRA is $(m + n)r$, which can be much smaller than $mn$ for small $r$. We can have a constraint on the number of parameters relative to the model size, i.e., $(m + n)r \leq \rho mn$, where $\rho > 0$ is the relative parameter budget per adaptor (e.g., $\rho = 0.01$ for a maximum of 1% increase in model size per adaptor). Given a specific $\rho$ based on system's capabilities, $r$ can be automatically set to be the maximum such that $r \leq \rho mn/(m + n)$, or just $r = \lfloor \rho mn/(m + n) \rfloor$. We hereafter refer to $\rho$ as the *budget* and set it to either 1% or 10% in the experiments. Note that, for certain models, it is impossible to satisfy the budget if $\rho mn/(m + n) < 1$, so we enforce a minimum rank of 1 as otherwise, there will be no adaptors.

**Convolution layer** Consider a 2D convolution layer. Let $\mathbf{W} \in \mathbb{R}^{c_{out} \times c_{in} \times k_1 \times k_2}$ be the base convolution layer. We similarly introduce a convolution "low-rank" adaptor (ConvLoRA) $\mathbf{L} = \mathbf{U} * \mathbf{V}$, such that the adaptove layer $\tilde{\mathbf{W}}$ becomes $(\mathbf{W} + \mathbf{L}) * x = \mathbf{W} * x + \mathbf{L} * x = \mathbf{W} * x + \mathbf{U} * (\mathbf{V} * x)$, where $*$ is the convolution operator. Note that ConvLoRA is introduced in the official implementation of [14], but it is a linear LoRA on a matricized convolution. In our case, we can have more than one way of defining $\mathbf{U}$ and $\mathbf{V}$. Depending on what is meant by "rank", we can either reduce the rank channel-wise, filter-wise, both, or as a linear layer by matricizing the convolution. We defer the full details to Appendix G.1, where we show that a novel channel+filter-wise implementation is more parameter-efficient and performs better. Further details about low-rank constructions of convolution layers can be found in [34, 35].

**Bias** Biases are vectors, so a low-rank parameterization would not be possible, and there is no straightforward way to have a parameter-efficient adaptor except by considering weight-sharing or a single constant. Due to biases contributing a small percentage of the overall number of parameters in large models, we consider adaptive biases as $\mathbf{b} + \mathbf{L_b}$ with extra biases $\mathbf{L_b}$ initialized to 0. Although this adaptor is not parameter-efficient relative to $\mathbf{b}$, the small impact on the overall parameter count means that this is not a significant limitation. Moreover, as demonstrated in Appendix H.3, this approach can be crucial for achieving optimal accuracy.

# 4. Analysis

In order to connect the analysis with our FLoRAL framework, we can consider a vector parameterization of the model given client $k$ and cluster $c$ as in (MFL-WS). Namely, we have $\mathbf{w}_{c,t}^k = (\mathbf{u}_t^k, \mathbf{a}_{c,t}^k)$, where it is understood as the concatenation of the two vectors and we emphasize that $\mathbf{u}_t^k$ does not depend on the cluster. For example, $\mathbf{u}_t^k = \text{vec}(\mathbf{W}_t^k)$ can be the base layer and $\mathbf{a}_{c,t}^k = (\text{vec}(\mathbf{U}_{c,t}^k)^\top \ \text{vec}(\mathbf{V}_{c,t}^k)^\top)^\top$ can be the LoRA adaptor. The analysis proceeds without assumptions on the form of $\mathbf{w}_{c,t}^k$. In Appendix C, we show the full analysis on (FML).

Recall $\boldsymbol{\pi}^k \in \Delta^{C-1}$ the ground-truth router of client $k$. In general, the probability of sampling a *single* client $k$ is often chosen to be proportional to the number of its data points, i.e., $p(k) \propto N^k$ (note this is different from sampling a *cohort*, which is explained below). On the other hand, the probability that client $k$ samples cluster $c$ is $p(c|k) = \boldsymbol{\pi}_c^k$ by construction. Since we have $p(k,c) = p(c|k)p(k) \propto \boldsymbol{\pi}_c^k N^k$, we can divide $p(k,c)$ by $p(c) = \sum_k p(k,c)$ to get $p(k|c)$. Overall, we have $p(c|k) = \boldsymbol{\pi}_c^k$ by construction and $p(k) = \frac{N^k}{N}$ by assumption, so that

$$ p(k,c) = \frac{N^k}{N} \boldsymbol{\pi}_c^k, \qquad p(c) = \sum_{k=1}^{K} \frac{N^k}{N} \boldsymbol{\pi}_c^k, \qquad p(k|c) = \frac{\boldsymbol{\pi}_c^k N^k}{\sum_{k'=1}^{K} \boldsymbol{\pi}_c^{k'} N^{k'}}. \tag{1} $$

**Notation** Denote $\hat{\boldsymbol{\pi}}_{c,t}^k$ the learned estimate of $\boldsymbol{\pi}_c^k$ at iteration $t$. Denote $\mathbf{p}_c^k := p(k|c) \propto \boldsymbol{\pi}_c^k N^k$ and similarly $\hat{\mathbf{p}}_{c,t}^k \propto \hat{\boldsymbol{\pi}}_{c,t}^k N^k$. Define the aggregation operators $\mathbb{E}_{k|c}[\mathbf{w}_{c,t}^k] := \sum_{k=1}^{K} \mathbf{p}_c^k \mathbf{w}_{c,t}^k$ and $\mathbb{E}_{c|k}[\mathbf{w}_{c,t}^k] := \sum_{c=1}^{C} \boldsymbol{\pi}_c^k \mathbf{w}_{c,t}^k$. Additionally, we denote using $\hat{\mathbb{E}}$ the same aggregation operators but taken with respect to $\hat{\mathbf{p}}_{c,t}^k$ and $\hat{\boldsymbol{\pi}}_{c,t}^k$, respectively.

Recall that the mixed (or personalized) objective of client $k$ is $\mathbb{E}_{c|k}[f^k(\mathbf{w}_{c,t}^k)] := \sum_{c=1}^{C} \boldsymbol{\pi}_c^k f^k(\mathbf{w}_{c,t}^k)$. The objective (MFL) can be stated more succinctly as

$$ \min_{\mathbf{w}_1, \cdots, \mathbf{w}_C} \quad \mathbb{E}_{c,k}[f^k(\mathbf{w}_c)]. \tag{2} $$

*Remark* 4.1. Consider a cluster assignment router (i.e., one-hot w.r.t. $c$). Let $k \sim \mathcal{K}$ and $\bar{c}$ be its associated cluster. Then, $\mathbb{E}_{c|k}[f^k(\mathbf{w}_c)] = f^k(\mathbf{w}_{\bar{c}})$ and $\mathbb{E}_{k|c}[f^k(\mathbf{w}_c)] = f_{\bar{c}}(\mathbf{w}_{\bar{c}})$.

**Local SGD** With the above notation in hand, we follow the local SGD framework with perturbed iterates [36]. Note that our work is orthogonal to [37] since they can estimate $p(k)$ with an unbiased participation indicator variable, whereas we assume that $p(k)$ is known and estimate $p(c|k)$ instead, which cannot be unbiased itself because of the dependency of the estimate on the optimal objective

values. Also, the analysis Pillutla et al. [10] cannot be directly adapted because it is concerned with a split of global and local variables (i.e., weights and mixture, respectively), whereas we take into account weight sharing across clusters and train mixtures (i.e., the local parameters) explicitly..

For client $k$ and cluster $c$, the algorithm starts with the initialization $\mathbf{w}_{c,0}^k = \hat{\mathbb{E}}_{k|c}[\mathbf{w}_{c,0}^k]$ with $\hat{\boldsymbol{\pi}}_{c,0}^k = 1/C$, without loss of generality. We define the aggregated gradient as $\mathbf{g}_{c,t}^k = \nabla f^{i_t}(\mathbf{w}_{c,t}^k)$ for independently sampled clients $i_t \sim \mathcal{K}$ every $H$ steps, i.e., $i_t = \cdots = i_{t_0}$ for all $t \geq t_0$ where $t_0 = t - (t \mod H)$. Though similar, we will explicitly reserve the random variables $i_t$ for denoting sampled clients at time $t$ and $k$ for denoting a "tracking" variable of the expected performance over clients, which will be independent of $i_t$. Let $c \in [C]$ and define $f_c := \mathbb{E}_{i_t|c}[f^i]$. Assume an unbiased estimate $\mathbb{E}_{i_t|c}[\mathbf{g}_{c,t}^k] = \nabla f_c(\mathbf{w}_{c,t}^k)$, where we denote $\mathbb{E}_{i_t|c}$ the expectation with respect to $i_t$ given $c$. Let $\mathbf{w}_c^*$ be any point satisfying $\nabla f_c(\mathbf{w}_c^*) = 0$. We run $T$ gradient steps $\mathbf{w}_{c,t+1}^k = \mathbf{w}_{c,t}^k - \eta_t \mathbf{g}_{c,t}^k$ with a learning rate $\eta_t$. Synchronization happens every $H$ iterations so that $\mathbf{w}_{c,t+1}^k = \hat{\mathbb{E}}_{k|c}[\mathbf{w}_{c,t}^k - \eta_t \mathbf{g}_{c,t}^k]$, $\forall t$ such that $(t+1) \mod H = 0$. The algorithm we use in the analysis is the following

$$\mathbf{w}_{c,t+1}^k = \begin{cases} \mathbf{w}_{c,t}^k - \eta_t \mathbf{g}_{c,t}^k, & \text{if } (t+1) \mod H > 0 \\ \hat{\mathbb{E}}_{k|c, \hat{\boldsymbol{\pi}}_t}[\mathbf{w}_{c,t}^k - \eta_t \mathbf{g}_{c,t}^k], & \text{otherwise,} \end{cases} \tag{3}$$

$$\hat{\boldsymbol{\pi}}_{t+1} \propto \begin{cases} \hat{\boldsymbol{\pi}}_t, & \text{if } (t+1) \mod H > 0 \\ \exp(-\eta_t f_c(\mathbf{w}_{c,t+1}^k)), & \text{otherwise.} \end{cases} \tag{4}$$

All of the practical implementation details will be discussed in more detail in the next section.

Following the local SGD analysis in [36], we make the following corresponding assumptions.

**Assumption 4.2** ($L$-smoothness and $\mu$-strong convexity). $f_c$ is $L$-smooth and $\mu$-strongly convex. In other words, $\forall \mathbf{w}, \mathbf{v} \in \mathbb{R}^d, \forall c$, the following holds

$$f_c(\mathbf{v}) - f_c(\mathbf{w}) - \langle \nabla f_c(\mathbf{w}), \mathbf{v} - \mathbf{w} \rangle \leq \frac{L}{2} \|\mathbf{v} - \mathbf{w}\|, \tag{5}$$

$$f_c(\mathbf{v}) - f_c(\mathbf{w}) - \langle \nabla f_c(\mathbf{w}), \mathbf{v} - \mathbf{w} \rangle \geq \frac{\mu}{2} \|\mathbf{v} - \mathbf{w}\|. \tag{6}$$

**Assumption 4.3** (Bounded second moment). $\forall \mathbf{w} \in \mathbb{R}^d, \forall c \in [C], \mathbb{E}_{i_t|c}\|\nabla f^{i_t}(\mathbf{w})\|^2 \leq G^2$.

**Assumption 4.4** (Bounded variance). $\forall \mathbf{w} \in \mathbb{R}^d, \forall c \in [C], \mathbb{E}_{i_t|c}\|\nabla f_c(\mathbf{w}) - \nabla f^{i_t}(\mathbf{w})\|^2 \leq \sigma^2$.

The main quantity of interest in our analysis is the total variation distance $\|\boldsymbol{\delta}_{c,t}\|_1$ where $\boldsymbol{\delta}_{c,t} := (|\hat{\mathbf{p}}_{c,t}^k - \mathbf{p}_c^k|)_{k=1}^K$. We may also refer to it as the *aggregation mismatch*, or just *mismatch*.

Using the router update in (4), we can obtain the convergence bound of local SGD but with an extra $\mathcal{O}(\frac{G}{\mu T})$ term and a learning rate inversely proportional to $\max\{L, G\}$ instead of $L$. This seems to be unavoidable without extra assumptions due to a circular dependency between $\boldsymbol{\delta}_{c,t}$ and $f_c(\mathbf{w}_{c,t}^k)$. However, we show in Corollary B.5 that local SGD descent is recovered when $\hat{\mathbf{p}}_{c,t}^k = \mathbf{p}_c^k$. The convergence rate for this general case can be seen in Theorem B.9.

Here, we present a convergence bound given an assumption on the decrease of $\|\boldsymbol{\delta}_{c,t}\|_1^2$. The exact bound can be found in Theorem B.10. We defer all proofs to Appendix B.

**Theorem 4.5.** *Consider the setup in Section 4. Let* $\tilde{\sigma}^2 = \sigma^2 \|\mathbf{p}_c\|^2$, $\kappa = \frac{L}{\mu}$, *and* $U_c = \min_{k; p(c) \leq \boldsymbol{\pi}_c^k}\{p(c)/\boldsymbol{\pi}_c^k\}$. *Initialize* $\hat{\boldsymbol{\pi}}_{c,0}^k = 1/C$ *for all* $k \in [K]$, *and assume* $|\mathbf{p}_c^k - \hat{\mathbf{p}}_{c,t}^k| \leq |\mathbf{p}_c^k - \hat{\mathbf{p}}_{c,0}^k|$ *for all* $t \geq 0$. *Assume that* $f_c(\mathbf{w}_c^*) = 0$ *without loss of generality, and assume that* $\|\boldsymbol{\delta}_{c,t}\|_1^2 \leq (t+s)^{-\beta}\|\boldsymbol{\delta}_{c,0}\|_1^2$ *for* $\beta \in (0,1)$. *Let* $\eta_t \leq \frac{\alpha}{t+s}$ *with* $\alpha = \frac{1}{\mu}$ *and* $s \geq \max\{3H, 4\kappa/U_c\}$. *Then,*

$$\mathbb{E}f_c(\hat{\mathbf{w}}_{c,T}) - f_c(\mathbf{w}_c^*) \leq \mathcal{O}\left(\frac{\tilde{\sigma}^2}{\mu T} + \frac{G^2\|\boldsymbol{\delta}_{c,0}\|_1^2}{\mu T^{1+\beta}} + \frac{G^2\kappa H^2}{\mu T^2}\right). \tag{7}$$

Observe that we recover local SGD asymptotically when $\|\boldsymbol{\delta}_{c,0}\|_1 = 0$ and $U_c = 1$ (which is the case for (FL)), or when $\beta \to 1$ since $\|\boldsymbol{\delta}_{c,0}\|_1 \leq 2$. Observe also that we obtain a general notion of variance reduction through $\tilde{\sigma}^2 = \sigma^2\|\mathbf{p}_c\|^2$. Indeed, $\|\mathbf{p}_c\|^2 = 1/K$ in the (FL) case and $\|\mathbf{p}_c\|^2 = 1/K_c$ for cluster $c$ in the (CFL) case, where $K_c$ is the number of clients in cluster $c$.

Note that $U_c \geq p(c) \approx 1/C$ for balanced clustered FL problems, but $p(c) \approx p(k)$ in the worst case when a cluster contains one client. The difficulty is inherent for such edge cases, but the dependence

on $U_c^{-1}$ in the bound appears only in higher-order terms (see Theorem B.10 for the full bound). We believe that having independent learning rates per client should remove the $\min$ in $U_c$, and a finer analysis on the quantity $\hat{\mathbf{p}}_{c,t}^k / \mathbf{p}_c^k$ can bound $U_c$ further from below, but we leave this for future work.

In Appendix C, we extend the analysis to the (FML) case with weight sharing (explained in the next section). Given fine-grained variances and cluster heterogeneity conditions for which weight sharing works best, we can demonstrate better variance reduction of the base layer under a trade-off with cluster heterogeneity (see (32), for example). A better understanding of weight sharing and the assumptions in Appendix C is an interesting direction for future work.

# 5. Practical Implementation

**Mixture of adaptors**  The (MFL) objective suggests that any learning algorithm will have to run at least $C$ forward passes per step for each client, which is necessary for computing the objective. One way to circumvent that is by "moving" the mixture inside the objective. This allows us to mix the weights and perform one forward pass. We call this Federated Mixture Learning (FML)[3]

$$
\min_{\{\mathbf{w}_c\}_{c=1}^C, \{\boldsymbol{\pi}^k\}_{k=1}^K} \quad \mathbb{E}_{k \sim \mathcal{K}} \left[ f^k \left( \sum_{c=1}^C \boldsymbol{\pi}_c^k \mathbf{w}_c \right) \right] + \Gamma(\{\mathbf{w}_c\}_{c=1}^C), \tag{FML}
$$
$$
\text{s.t.} \quad \boldsymbol{\pi}^k \in \Delta^{C-1}, \forall k \in [K].
$$

Observe that for convex $f^k$, this proxy acts as a lower bound since $f^k(\sum_{c=1}^C \boldsymbol{\pi}_c^k \mathbf{w}_c) \leq \sum_{c=1}^C \boldsymbol{\pi}_c^k f^k(\mathbf{w}_c)$ due to Jensen's inequality. Thus, for convex losses $f^k$, minimizing (MFL) implies minimizing (FML), but not vice versa. In this sense, (FML) could be seen as a more general problem, and (MFL) is a relaxation. We note that this problem is similar to FedEM [3], but we only use $K$ mixture vectors of size $C$ and we do not have sample-specific weights.

This formulation is especially useful for additive adaptors since the weights can be merged into one. Also, it allows us to mix the $C$ adaptors and run one forward pass, which is often more efficient than running $C$ forward passes. This is particularly true for inference, in which the weights can be merged once so that forward passes come without extra cost. The benefits of weight sharing can also manifest through better variance reduction, which is demonstrated in Appendix C.

**Learning the mixture weights**  Instead of optimizing $\boldsymbol{\pi}^k$ directly in $\Delta^{C-1}$, we consider the parameterization $\boldsymbol{\pi}^k = \text{Softmax}(\boldsymbol{\theta}^k)$ for some vector $\boldsymbol{\theta}^k \in \mathbb{R}^C$. Note that $\boldsymbol{\theta}^k$ is a local parameter and not aggregated. The cost of storing $\boldsymbol{\theta}^k$ in each client is minimal as it is of size $C$, which is often significantly small compared to the model size $d$. Even if we consider stateless clients, the server should be able to handle an extra storage and communication budget of $\boldsymbol{\theta}^k$, which is $KC$. Note that the server does not need to know the IDs of the clients and that the clients can learn the $\boldsymbol{\theta}^k$ from scratch every round, as it is not expensive. Let us consider a scenario where the cost $KC$ is prohibitive. Suppose the model size is $d = 1000$ and the client participation ratio is $p = 0.1\%$. The extra cost for the server will be $pKd = K < KC$ for $C > 1$. Thus, the prohibitive scenario occurs only when $pd < C$, which is often not the case as $d$ is rarely this small (e.g., a 32 by 32 linear layer with bias has more 1000 parameters), let alone $p$. The only drawback with stateless clients is the need to learn $\boldsymbol{\theta}^k$ from scratch every round, which is cheap to learn given the current model.

In Appendix D, we make a connection between the router update in (4) for (MFL) and the gradient descent update of $\boldsymbol{\pi}^k$ on (FML) under the Softmax parameterization, and show conditions under which they become equivalent.

**FLoRAL problem and algorithm**  We obtain the FLoRAL problem by employing the weight sharing regularizer in (MFL-WS) to (FML) and using low-rank adaptors $\mathbf{a}_c$. Weight sharing and low-rankedness are explicit in the parameterization. The algorithm we use to solve (FLoRAL) in practice is shown in Appendix A and is straightforward. We use simultaneous gradient descent for $\mathbf{u}$ and

---

[3]The "M" in the acronym follows the position of the mixture in the objective.

$\mathbf{a}_c$, so we simply write the update in terms of the concatenation $\mathbf{w}_c$. One trick we employ to ensure better convergence is LoRA preconditioning, which is discussed in Appendix E.

# 6. Experiments

In this section, we compare FLoRAL with 3 methods: (i) FedAvg, which uses the base model only without adaptors, (ii) Local Adaptor, which uses an adaptor for *each* client, and (iii) Ensemble, which uses a mixture of $C$ copies of the base model. The datasets considered have known ground-truth clusters and are inspired from [4, 19]. Further, we test on the same datasets with only 95% of each client's data dropped. This is to demonstrate the benefits of our parsimonious parameterization, where a large model such as Ensemble might overfit on the local datasets. The results can be seen in Table 1. Further ablation studies on $\rho$ and $C$, the adaptors, and the type of ConvLoRAs can be found in Table 2, Table 3, and Table 4, respectively.

In general, we follow the experimental setup in [4] or [10] and implement our experiments using PyTorch [17] and Flower [18]. We use the simplest setup possible without any tricks other than LoRA preconditioning, which is explained in Appendix E. We discuss another trick called LoRA centering in Appendix F, which we believe is potentially useful. The algorithm we use in practice is shown in Algorithm 1. Further details can be found in Appendix H.

Table 1: Accuracy of different methods on our tasks. $\pi^*$ indicates the use of optimal routing. Full = 100% data, Reduced = 5% data. R = Rotate, LS = Label Shift. **Bold** = best, *italic* = second best.

| Method | $\pi^*$ | MNIST | | | | CIFAR-10 | | | | CIFAR-100 | |
|---|---|---|---|---|---|---|---|---|---|---|---|
| | | Full | | Reduced | | Full | | Reduced | | Full | Reduced |
| | | R | LS | R | LS | R | LS | R | LS | | |
| FedAvg | | 91.5$_{0.6}$ | 25.8$_{2.4}$ | 78.2$_{0.6}$ | 23.2$_{0.9}$ | 64.4$_{0.3}$ | 21.9$_{0.4}$ | 45.6$_{0.3}$ | 18.7$_{0.4}$ | 29.2$_{1.8}$ | 20.7$_{1.4}$ |
| Local Adaptor | | 86.6$_{0.3}$ | 84.5$_{1.8}$ | 47.4$_{5.4}$ | 32.0$_{2.3}$ | 66.3$_{0.5}$ | 68.8$_{0.5}$ | 33.5$_{0.5}$ | 30.8$_{0.8}$ | 85.1$_{0.8}$ | 39.5$_{2.8}$ |
| Ensemble | ✗ | 92.0$_{0.1}$ | 93.8$_{0.5}$ | 66.7$_{5.3}$ | 86.4$_{0.4}$ | 71.0$_{2.8}$ | 46.4$_{9.2}$ | 42.4$_{0.9}$ | 41.7$_{4.6}$ | 86.2$_{0.0}$ | 43.7$_{3.2}$ |
| Ensemble | ✓ | **95.8**$_{0.3}$ | **95.6**$_{0.5}$ | **88.2**$_{1.4}$ | **87.6**$_{1.3}$ | **73.7**$_{0.2}$ | **73.3**$_{0.1}$ | 45.0$_{0.9}$ | **45.1**$_{0.8}$ | **92.8**$_{0.3}$ | **55.0**$_{0.4}$ |
| FLoRAL(1%) | ✗ | 91.3$_{0.6}$ | 89.7$_{3.2}$ | 73.1$_{3.7}$ | 46.0$_{9.9}$ | 65.5$_{0.4}$ | 62.8$_{8.8}$ | 45.2$_{0.3}$ | 44.2$_{0.9}$ | 81.3$_{0.5}$ | 52.2$_{0.5}$ |
| FLoRAL(1%) | ✓ | 93.9$_{0.8}$ | 93.7$_{0.2}$ | *87.5*$_{2.1}$ | **87.6**$_{0.5}$ | 68.9$_{0.2}$ | 72.2$_{0.2}$ | **47.8**$_{0.9}$ | 44.1$_{0.6}$ | 82.4$_{0.2}$ | 53.1$_{0.4}$ |
| FLoRAL(10%) | ✗ | 91.8$_{1.0}$ | 93.1$_{0.9}$ | 75.7$_{2.3}$ | 70.8$_{7.1}$ | 65.1$_{0.3}$ | 56.2$_{5.5}$ | 44.5$_{0.4}$ | 42.1$_{0.2}$ | *87.3*$_{0.3}$ | 51.2$_{1.0}$ |
| FLoRAL(10%) | ✓ | *94.5*$_{0.6}$ | *94.2*$_{0.2}$ | 87.0$_{0.7}$ | *86.9*$_{0.5}$ | *69.3*$_{0.5}$ | *72.1*$_{0.5}$ | *47.2*$_{0.3}$ | *42.7*$_{0.3}$ | 86.6$_{0.5}$ | *53.9*$_{0.9}$ |

**Synthetic** Consider a regression task where we want to learn $\mathbf{y} \in \mathbb{R}^{d_y}$ given $\mathbf{x} \in \mathbb{R}^{d_x}$, where $\mathbf{x} \sim \mathcal{N}(0, \mathbf{I}_{d_x})$. We construct two versions of this regression task: one is based on a linear model plus a personalized LoRA, and the other is based on a similar setup on the first layer of a two-layer ReLU net. Namely, the target model for client $k$ is

$$\mathbf{y}_{\text{lin}}^k(\mathbf{x}) = \sum_{c=1}^{C} \boldsymbol{\pi}_c^k (\mathbf{W} + \alpha \mathbf{U}_c \mathbf{V}_c^\top)\mathbf{x}, \tag{8}$$

where $\mathbf{W} \in \mathbb{R}^{d_y \times d_x}$, $\mathbf{U}_c \in \mathbb{R}^{d_y \times r}$, $\mathbf{V}_c \in \mathbb{R}^{d_x \times r}$, and $\alpha \in \mathbb{R}$. Similarly, consider the 2-layer ReLU neural net $\mathbf{y}_{\text{mlp}}^k(\mathbf{x}) = \boldsymbol{\Phi}(\mathbf{y}_{\text{lin}}^k(\mathbf{x}))_+$ for $\boldsymbol{\Phi} \in \mathbb{R}^{d_y \times d_y}$, where we write the ReLU function as $(\cdot)_+$. These tasks provide a *proof of concept* for FLoRAL. We discuss these datasets in more detail in Appendix H.1. The results in Figure 2 show the performances with $K = 10$ and $C = 2$ for the linear version and $K = 20$ and $C = 4$ for the MLP version. Note that even the linear task is not easy to solve, and similar problems have been studied in the mixed linear regression literature, e.g., see [38].

**MNIST and CIFAR-10** We test our method on a clustered version of MNIST and CIFAR-10 datasets in which the clusters are generated according to one of the following tasks: 1) a rotation task, where each cluster $c$ rotates the image by $2\pi c/C$ degrees, and 2) a label shift task, where cluster $c$ shifts the labels by $y \mapsto (y + c) \mod 10$. Following [4], we choose $C = 4$ and $K = 300$ for MNIST and sample 10% of the clients every round, and choose $C = 4$ and $K = 20$ for CIFAR-10 and sample all clients every round. The model for MNIST is a 2-layer ReLU net, whereas for CIFAR-10, it has two convolution layers followed by a 2-layer ReLU net classifier.

Table 2: Ablation of $\rho$ and $C$.

| $C$ | $\rho$ | CIFAR-10 R | CIFAR-10 LS | CIFAR-100 |
|---|---|---|---|---|
| $\times 0.5$ | 1% | 66.5 | 36.3 | 48.8 |
| $\times 0.5$ | 10% | 66.8 | 41.6 | 50.9 |
| $\times 1$ | 1% | 70.2 | *74.1* | 51.7 |
| $\times 1$ | 10% | **71.5** | **74.2** | **57.4** |
| $\times 2$ | 1% | 69.0 | 73.8 | 51.3 |
| $\times 2$ | 10% | *70.8* | *74.1* | *54.8* |

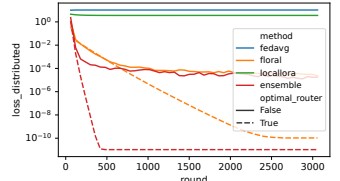 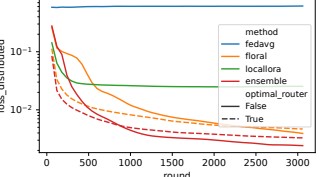

Figure 2: Test loss on linear and MLP synthetic datasets.

**CIFAR-100** The CIFAR-100 task is to train a model that is not expressive enough to fit 100 labels yet expressive enough to fit 10 labels. Thus, we expect that the model would benefit from collaboration with the right clients. The setup is to divide the 100 labels into $C = 10$ clusters such that each cluster has 10 unique labels and then split each cluster uniformly into $K/C = 10$ clients (so, in total, $K = 100$). The model used is VGG-8, a custom-sized model from the VGG-family [39] that is specifically able to fit 10 labels but not 100. We sample 25 clients every round, which makes the task harder than [4] and can result in overfitting.

**Discussion** The results in Figure 9 show the robustness of FLoRAL, particularly when $C$ is larger than the number of ground-truth clusters. In Table 1, we can see that FLoRAL is always competitive with the best baseline, which is Ensemble given optimal routers. A particularly interesting case is the reduced CIFAR-10-R experiments, in which FLoRAL(1%) and FLoRAL(10%) surprisingly outperform this baseline, even *in the optimal routing case*. This seems slightly counter-intuitive as Ensemble is strictly more expressive than FLoRAL. We believe this to be due to the variance reduction shown in Appendix C.

Note that FLoRAL($\rho$) has $C\rho d$ extra parameters, whereas Ensemble has $(C-1)d$. For example, when $d = 1,000$ and $C = 4$, FLoRAL(1%) adds 40 parameters vs. $3,000$ for Ensemble, and when $C = 10$, it is 100 vs. $9,000$. Local Adaptor requires each client to have its own adaptor (i.e., each client has $\rho d$ memory). Regardless of its feasibility, FLoRAL is shown to leverage the power of collaboration when Local Adaptor fail to do so. The low accuracies of FLoRAL with learned routing in the reduced MNIST-LS can be alleviated with more training rounds, e.g., see Appendix H.5 for full plots.

Overall, the results demonstrate that FLoRAL is a collaborative and efficient personalization method, and it can lead to better generalization in low-data regimes.

## 7. Conclusion

In this work, we presented a parameter-efficient method for collaborative learning and personalization. Here are some future directions we are interested in exploring:

- Is there a principled way to understand the trade-off between parameter-efficiency and the accuracy gains from increasing $\rho$ or $C$ and how to choose them in practice?
- (FML) can be formulated as a "multimodal optimization" problem [40], or a model class of mixture-candidate distributions [41]. Can we design more efficient algorithms under this framework with a mixture of structured distributions [42]?
- Would FLoRAL be suitable for federated fine-tuning of language models?
- The router $\pi$ can route based on its input, as in mixture of experts [43]. It can also be learned per layer. Preliminary experiments show marginal benefits, but there is still room for exploration.
- We are interested in designing methods for zero-shot generalization to unseen clients based on FLoRAL. Is it possible to fine-tune the router without labels?

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

## A. Algorithm

We show in this section a simplified version of the algorithm we use in practice. The algorithm is straightforward gradient descent. The only different part is the parameterization of the FLoRAL model. Note that $\theta^k \in \mathbb{R}^C$ are local parameters or can be learned from scratch every round by (4).

---

**Algorithm 1** Simple FLoRAL Averaging

---

1: Let $\mathbf{w}_{c,t}^k = (\mathbf{u}_t^k, \mathbf{a}_{c,t}^k)$
2: **for** $\tau = 0, H, 2H, \cdots, \lfloor \frac{T-1}{H} \rfloor$ **do**          ▷ Comm. rounds
3:     Sample clients $S_\tau \sim \mathcal{K}$
4:     **for all** $k \in S_\tau$ **in parallel do**
5:        **for** $t = \tau, \cdots, \tau + H - 1$ **do**          ▷ Local epoch
6:           $\hat{\boldsymbol{\pi}}_{c,t}^k = \exp(\theta_{c,t}^k) / \sum_{c=1}^C \exp(\theta_{c,t}^k)$
7:           $\theta_{c,t+1}^k = \theta_{c,t}^k - \eta_t \nabla_{\theta_{c,t}^k} f^k (\sum_{c=1}^C \hat{\boldsymbol{\pi}}_{c,t}^k \mathbf{w}_{c,t}^k)$
8:           $\mathbf{w}_{c,t+1}^k = \mathbf{w}_{c,t}^k - \eta_t \nabla_{\mathbf{w}_{c,t}^k} f^k (\sum_{c=1}^C \hat{\boldsymbol{\pi}}_{c,t}^k \mathbf{w}_{c,t}^k)$
9:        **end for**
10:    **end for**
11:    $\mathbf{u}_{\tau+H}^k \leftarrow \frac{\sum_{k \in S_\tau} N^k \mathbf{u}_{\tau+H}^k}{\sum_{k \in S_\tau} N^k}$          ▷ Synchronize base layers
12:    $\mathbf{a}_{c,\tau+H}^k \leftarrow \frac{\sum_{k \in S_\tau} \hat{\boldsymbol{\pi}}_{c,\tau+H}^k N^k \mathbf{a}_{c,\tau+H}^k}{\sum_{k \in S_\tau} \hat{\boldsymbol{\pi}}_{c,\tau+H}^k N^k}$          ▷ Synchronize adaptors
13: **end for**

---

## B. Proofs

We reiterate the notations part from the main text here for clarity. Let $\mathbf{p}_c^k := \frac{\boldsymbol{\pi}_c^k N^k}{\sum_{k'=1}^K \boldsymbol{\pi}_c^{k'} N_{k'}} = p(k|c)$ and $\hat{\mathbf{p}}_{c,t}^k := \frac{\hat{\boldsymbol{\pi}}_{c,t}^k N^k}{\sum_{k'=1}^K \hat{\boldsymbol{\pi}}_{c,t}^{k'} N_{k'}}$. Define the expectation operators $\mathbb{E}_{k|c}[\mathbf{w}_{c,t}^k] := \sum_{k=1}^K \mathbf{p}_c^k \mathbf{w}_{c,t}^k$ and $\mathbb{E}_{c|k}[\mathbf{w}_{c,t}^k] := \sum_{c=1}^C \boldsymbol{\pi}_c^k \mathbf{w}_{c,t}^k$ and similarly for their estimates $\hat{\mathbb{E}}_{k|c,\hat{\boldsymbol{\pi}}_t}[\mathbf{w}_{c,t}^k]$ and $\hat{\mathbb{E}}_{c|k,\hat{\boldsymbol{\pi}}_t}[\mathbf{w}_{c,t}^k]$. We drop $\hat{\boldsymbol{\pi}}_{c,t}^k$ from the notation for clarity. We use the variable $i$ to denote client sampling, and $i|c$ should be understood as randomness in client sampling given cluster $c$, for example. Finally, let the global function of cluster $c$ be $f_c(\mathbf{w}) := \mathbb{E}_{k|c}[f^k(\mathbf{w})]$. Note the absence of $k$ in the weight.

The analysis roughly follows [36] and differ mostly in the appearance of the total variation distance between $\mathbf{p}_c^k$ and $\hat{\mathbf{p}}_{c,t}^k$.

We start by introducing virtual iterates for tracking the aggregated weights (or gradients) with respect to the true router (or the estimated router) at every time step, which will be mainly useful for the analysis. These iterates coincide at the synchronization step, in which they become equal by construction of the algorithm. The iterates are as follows

$$\tilde{\mathbf{w}}_{c,t} := \hat{\mathbb{E}}_{k|c}[\mathbf{w}_{c,t}^k], \qquad \tilde{\mathbf{g}}_{c,t} := \hat{\mathbb{E}}_{k|c}[\nabla f^{i_t}(\mathbf{w}_{c,t}^k)], \tag{9}$$

$$\bar{\mathbf{w}}_{c,t} := \mathbb{E}_{k|c}[\mathbf{w}_{c,t}^k], \qquad \bar{\mathbf{g}}_{c,t} := \mathbb{E}_{k|c}[\nabla f_c(\mathbf{w}_{c,t}^k)], \tag{10}$$

Note that $\tilde{\mathbf{w}}_{c,t+1} = \tilde{\mathbf{w}}_{c,t} - \eta_t \tilde{\mathbf{g}}_{c,t}$ and $\mathbb{E}_{i_t|c}[\tilde{\mathbf{g}}_{c,t}] = \bar{\mathbf{g}}_{c,t}$. Hence, using $\|\mathbf{a} + \mathbf{b}\|^2 \leq 2\|\mathbf{a}\|^2 + 2\|\mathbf{b}\|^2$, we have

$$\mathbb{E}_{i_t|c}\|\tilde{\mathbf{w}}_{c,t+1} - \mathbf{w}_c^*\|^2 = \mathbb{E}_{i_t|c}\|\tilde{\mathbf{w}}_{c,t} - \mathbf{w}_c^* - \eta_t \tilde{\mathbf{g}}_{c,t}\|^2$$

$$= \mathbb{E}_{i_t|c}\|\tilde{\mathbf{w}}_{c,t} - \mathbf{w}_c^* - \eta_t \tilde{\mathbf{g}}_{c,t} - \eta_t \bar{\mathbf{g}}_{c,t} + \eta_t \bar{\mathbf{g}}_{c,t}\|^2$$

$$= \underbrace{\mathbb{E}_{i_t|c}\|\tilde{\mathbf{w}}_{c,t} - \mathbf{w}_c^* - \eta_t \bar{\mathbf{g}}_{c,t}\|^2}_{\text{ideal aggregation descent}} + \eta_t^2 \underbrace{\mathbb{E}_{i_t|c}\|\bar{\mathbf{g}}_{c,t} - \tilde{\mathbf{g}}_{c,t}\|^2}_{\text{gradient aggregation error}}$$

$$+ 2\eta_t \underbrace{\mathbb{E}_{i_t|c}\langle \tilde{\mathbf{w}}_{c,t} - \mathbf{w}_c^* - \eta_t \bar{\mathbf{g}}_{c,t}, \bar{\mathbf{g}}_{c,t} - \tilde{\mathbf{g}}_{c,t}\rangle}_{\text{correlation error}} . \tag{11}$$

In the original local SGD analysis, the correlation error is 0 since we aggregate the sampled gradients exactly and thus the expectation gives $\mathbb{E}_{i_t|c}[\tilde{\mathbf{g}}_{c,t}] = \bar{\mathbf{g}}_{c,t}$. Note that the expectation $\mathbb{E}_{i_t|c}$ is implicitly defined $\mathbb{E}_{i_t|c}[\cdot|i_{t-1},\cdots]$, which would be $\mathbb{E}_{i_t|c}[\cdot|i_{t_0-1},\cdots]$, where $t_0 = t - (t \mod H)$ since $i_t = \cdots = i_{t_0}$ (because we sample clients every $H$ round).

## B.1. Bounding descent

**Lemma B.1** (Descent bound 1). *Given the setting and the assumptions in Section 4, the following holds*

$$\|\tilde{\mathbf{w}}_{c,t+1} - \mathbf{w}_c^*\|^2 \leq (1 - \eta_t\mu)\|\tilde{\mathbf{w}}_{c,t} - \mathbf{w}_c^*\|^2 + 2\eta_t^2\|\tilde{\mathbf{g}}_{c,t} - \bar{\mathbf{g}}_{c,t}\|^2 + 2L\eta_t\hat{\mathbb{E}}_{k|c}\|\tilde{\mathbf{w}}_{c,t} - \mathbf{w}_{c,t}^k\|^2$$

$$+ \eta_t\sum_{k=1}^{K}(4L\eta_t\mathbf{p}_c^k - \hat{\mathbf{p}}_{c,t}^k)[f_c(\mathbf{w}_{c,t}^k) - f_c(\mathbf{w}_c^*)]$$

*Proof.* From the ideal aggregation descent, we have

$$\|\tilde{\mathbf{w}}_{c,t} - \mathbf{w}_c^* - \eta_t\bar{\mathbf{g}}_{c,t}\|^2 = \|\tilde{\mathbf{w}}_{c,t} - \mathbf{w}_c^*\|^2 + \eta_t^2\|\bar{\mathbf{g}}_{c,t}\|^2 - 2\eta_t\langle\tilde{\mathbf{w}}_{c,t} - \mathbf{w}_c^*, \bar{\mathbf{g}}_{c,t}\rangle$$

$$\leq \|\tilde{\mathbf{w}}_{c,t} - \mathbf{w}_c^*\|^2 + \eta_t^2\mathbb{E}_{k|c}\|\nabla f_c(\mathbf{w}_{c,t}^k)\|^2 - 2\eta_t\langle\tilde{\mathbf{w}}_{c,t} - \mathbf{w}_c^*, \bar{\mathbf{g}}_{c,t}\rangle,$$

where we have used Jensen's inequality [4]. As for the correlation error, we can write it as

$$2\eta_t\mathbb{E}_{i_t|c}\langle\tilde{\mathbf{w}}_{c,t} - \mathbf{w}_c^* - \eta_t\bar{\mathbf{g}}_{c,t}, \bar{\mathbf{g}}_{c,t} - \tilde{\mathbf{g}}_{c,t}\rangle = 2\eta_t\langle\tilde{\mathbf{w}}_{c,t} - \mathbf{w}_c^*, \bar{\mathbf{g}}_{c,t} - \tilde{\mathbf{g}}_{c,t}\rangle - 2\eta_t^2\langle\bar{\mathbf{g}}_{c,t}, \bar{\mathbf{g}}_{c,t} - \tilde{\mathbf{g}}_{c,t}\rangle.$$

We bound $-\eta_t^2\langle\bar{\mathbf{g}}_{c,t}, \bar{\mathbf{g}}_{c,t} - \tilde{\mathbf{g}}_{c,t}\rangle$ with Young's inequality [5]

$$-2\eta_t^2\langle\bar{\mathbf{g}}_{c,t}, \bar{\mathbf{g}}_{c,t} - \tilde{\mathbf{g}}_{c,t}\rangle \leq \eta_t^2\|\bar{\mathbf{g}}_{c,t}\|^2 + \eta_t^2\|\bar{\mathbf{g}}_{c,t} - \tilde{\mathbf{g}}_{c,t}\|^2$$

$$\leq \eta_t^2\mathbb{E}_{k|c}\|\nabla f_c(\mathbf{w}_{c,t}^k)\|^2 + \eta_t^2\|\bar{\mathbf{g}}_{c,t} - \tilde{\mathbf{g}}_{c,t}\|^2,$$

where we have used Jensen's inequality as before.

Adding everything together, we get

$$\|\tilde{\mathbf{w}}_{c,t+1} - \mathbf{w}_c^*\|^2 \leq \|\tilde{\mathbf{w}}_{c,t} - \mathbf{w}_c^*\|^2 + 2\eta_t^2\mathbb{E}_{k|c}\|\nabla f_c(\mathbf{w}_{c,t}^k)\|^2 + 2\eta_t^2\|\tilde{\mathbf{g}}_{c,t} - \bar{\mathbf{g}}_{c,t}\|^2$$

$$- 2\eta_t\hat{\mathbb{E}}_{k|c}\langle\tilde{\mathbf{w}}_{c,t} - \mathbf{w}_c^*, \nabla f_c(\mathbf{w}_{c,t}^k)\rangle$$

$$= \|\tilde{\mathbf{w}}_{c,t} - \mathbf{w}_c^*\|^2 + 2\eta_t^2\mathbb{E}_{k|c}\|\nabla f_c(\mathbf{w}_{c,t}^k)\|^2 + 2\eta_t^2\|\tilde{\mathbf{g}}_{c,t} - \bar{\mathbf{g}}_{c,t}\|^2$$

$$- 2\eta_t\hat{\mathbb{E}}_{k|c}\langle\tilde{\mathbf{w}}_{c,t} - \mathbf{w}_{c,t}^k, \nabla f_c(\mathbf{w}_{c,t}^k)\rangle - 2\eta_t\hat{\mathbb{E}}_{k|c}\langle\mathbf{w}_{c,t}^k - \mathbf{w}_c^*, \nabla f_c(\mathbf{w}_{c,t}^k)\rangle.$$

Observe that, by Assumption 4.2 and $\nabla f_c(\mathbf{w}_c^*) = \mathbf{0}$, we have

$$\|\nabla f_c(\mathbf{w}_{c,t}^k)\|^2 = \|\nabla f_c(\mathbf{w}_{c,t}^k) - \nabla f_c(\mathbf{w}_c^*)\|^2 \leq 2L[f_c(\mathbf{w}_{c,t}^k) - f_c(\mathbf{w}_c^*)], \qquad (12)$$

and

$$-\langle\mathbf{w}_{c,t}^k - \mathbf{w}_c^*, \nabla f_c(\mathbf{w}_{c,t}^k)\rangle \leq -[f_c(\mathbf{w}_{c,t}^k) - f_c(\mathbf{w}_c^*)] - \frac{\mu}{2}\|\mathbf{w}_{c,t}^k - \mathbf{w}_c^*\|^2. \qquad (13)$$

We bound $\langle\tilde{\mathbf{w}}_{c,t} - \mathbf{w}_{c,t}^k, \nabla f_c(\mathbf{w}_{c,t}^k)\rangle$ with Young's inequality

$$-2\langle\tilde{\mathbf{w}}_{c,t} - \mathbf{w}_{c,t}^k, \nabla f_c(\mathbf{w}_{c,t}^k)\rangle \leq 2L\|\tilde{\mathbf{w}}_{c,t} - \mathbf{w}_{c,t}^k\|^2 + \frac{1}{2L}\|\nabla f_c(\mathbf{w}_{c,t}^k)\|^2$$

$$\overset{(12)}{\leq} 2L\|\tilde{\mathbf{w}}_{c,t} - \mathbf{w}_{c,t}^k\|^2 + [f_c(\mathbf{w}_{c,t}^k) - f_c(\mathbf{w}_c^*)].$$

We now plug in the results into the main bound

$$\|\tilde{\mathbf{w}}_{c,t+1} - \mathbf{w}_c^*\|^2 \leq \|\tilde{\mathbf{w}}_{c,t} - \mathbf{w}_c^*\|^2 + 4L\eta_t^2\mathbb{E}_{k|c}[f_c(\mathbf{w}_{c,t}^k) - f_c(\mathbf{w}_c^*)] + 2\eta_t^2\|\tilde{\mathbf{g}}_{c,t} - \bar{\mathbf{g}}_{c,t}\|^2$$

$$+ 2\eta_t L\hat{\mathbb{E}}_{k|c}\|\tilde{\mathbf{w}}_{c,t} - \mathbf{w}_{c,t}^k\|^2 + \eta_t\hat{\mathbb{E}}_{k|c}[f_c(\mathbf{w}_{c,t}^k) - f_c(\mathbf{w}_c^*)])$$

$$- 2\eta_t\hat{\mathbb{E}}_{k|c}[f_c(\mathbf{w}_{c,t}^k) - f_c(\mathbf{w}_c^*)] - \eta_t\mu\hat{\mathbb{E}}_{k|c}\|\mathbf{w}_{c,t}^k - \mathbf{w}_c^*\|^2$$

$$\leq (1 - \eta_t\mu)\|\tilde{\mathbf{w}}_{c,t} - \mathbf{w}_c^*\|^2 + 2\eta_t^2\|\tilde{\mathbf{g}}_{c,t} - \bar{\mathbf{g}}_{c,t}\|^2 + 2L\eta_t\hat{\mathbb{E}}_{k|c}\|\tilde{\mathbf{w}}_{c,t} - \mathbf{w}_{c,t}^k\|^2$$

$$+ \eta_t\sum_{k=1}^{K}(4L\eta_t\mathbf{p}_c^k - \hat{\mathbf{p}}_{c,t}^k)[f_c(\mathbf{w}_{c,t}^k) - f_c(\mathbf{w}_c^*)],$$

---

[4] $f(\mathbb{E}X) \leq \mathbb{E}f(X)$ for random variable $X$ and convex $f$.
[5] $2\langle\mathbf{a}, \mathbf{b}\rangle \leq \gamma\|\mathbf{a}\|^2 + \gamma^{-1}\|\mathbf{b}\|^2$ for $\gamma > 0$.

where we have used Jensen's inequality $-\hat{\mathbb{E}}_{k|c}\|\mathbf{w}_{c,t}^k - \mathbf{w}_c^*\|^2 \leq -\|\tilde{\mathbf{w}}_{c,t} - \mathbf{w}_c^*\|^2$. This completes the proof.

$\square$

**Lemma B.2** (Gradient aggregation error). *Let $\boldsymbol{\delta}_{c,t}^k := |\hat{\mathbf{p}}_{c,t}^k - \mathbf{p}_c^k|$ and $\boldsymbol{\delta}_{c,t} := (\boldsymbol{\delta}_{c,t}^k)_{k=1}^K$. Then,*

$$\mathbb{E}_{i_t|c}\|\bar{\mathbf{g}}_{c,t} - \tilde{\mathbf{g}}_{c,t}\|^2 \leq 2\sigma^2\|\mathbf{p}_c\|^2 + 2G^2\|\boldsymbol{\delta}_{c,t}\|_1^2. \tag{14}$$

*Proof.* We divide the gradient aggregation error into controllable terms.

$$\|\bar{\mathbf{g}}_{c,t} - \tilde{\mathbf{g}}_{c,t}\|^2 = \|\sum_{k=1}^K \mathbf{p}_c^k \nabla f_c(\mathbf{w}_{c,t}^k) - \hat{\mathbf{p}}_{c,t}^k \nabla f^{i_t}(\mathbf{w}_{c,t}^k)\|^2$$

$$= \|\sum_{k=1}^K \mathbf{p}_c^k(\nabla f_c(\mathbf{w}_{c,t}^k) - \nabla f^{i_t}(\mathbf{w}_{c,t}^k)) + (\mathbf{p}_c^k - \hat{\mathbf{p}}_{c,t}^k)\nabla f^{i_t}(\mathbf{w}_{c,t}^k)\|^2$$

$$\leq 2\|\sum_{k=1}^K \mathbf{p}_c^k(\nabla f_c(\mathbf{w}_{c,t}^k) - \nabla f^{i_t}(\mathbf{w}_{c,t}^k))\|^2 + 2\|\sum_{k=1}^K \boldsymbol{\delta}_{c,t}^k \nabla f^{i_t}(\mathbf{w}_{c,t}^k)\|^2. \tag{15}$$

The first term can be bounded by noting $\text{Var}(\sum_{k=1}^K c_k X_k) = \sum_{k=1}^K c_k^2 \text{Var}(X_k)$ for independent $X_k$, which holds since we condition on the previous iterates. We use Assumption 4.4 to obtain

$$\mathbb{E}_{i_t|c}\|\sum_{k=1}^K \mathbf{p}_c^k(\nabla f_c(\mathbf{w}_{c,t}^k) - \nabla f^{i_t}(\mathbf{w}_{c,t}^k))\|^2 \leq \sum_{k=1}^K (\mathbf{p}_c^k)^2 \text{Var}(\nabla f^{i_t}(\mathbf{w}_{c,t}^k)) \leq \sigma^2\|\mathbf{p}_c\|^2. \tag{16}$$

The second term can be bounded with Jensen's inequality and Assumption 4.3. Note $i_t = \cdots = i_{t_0}$ for $t_0 = t - (t \mod H)$ and $\boldsymbol{\delta}_{c,t}^k$ does not depend on $i_t$ for $t \geq t_0$ by construction, as shown in (4), so

$$\mathbb{E}_{i_t|c}\|\sum_{k=1}^K \boldsymbol{\delta}_{c,t}^k \nabla f^{i_t}(\mathbf{w}_{c,t}^k)\|^2 \leq \|\boldsymbol{\delta}_{c,t}\|_1 \sum_{k=1}^K \boldsymbol{\delta}_{c,t}^k \mathbb{E}_{i_t|c}\|\nabla f^{i_t}(\mathbf{w}_{c,t}^k)\|^2 \leq \|\boldsymbol{\delta}_{c,t}\|_1^2 G^2. \tag{17}$$

Combining (16) and (17) into (15) and taking expectation completes the proof. $\square$

**Lemma B.3** (Weights second moment). *Assume that $\eta_{t+1} \leq \eta_t$ and $\eta_{t_0} \leq 2\eta_t$, where $t_0 = t - (t \mod H)$, i.e., $\eta_t \leq \eta_{t_0} \leq 2\eta_t$. Then, we have*

$$\mathbb{E}_{i_t|c}\mathbb{E}_{k|c}\|\bar{\mathbf{w}}_{c,t} - \mathbf{w}_{c,t}^k\|^2 \leq 4\eta_t^2 H^2 G^2,$$

$$\mathbb{E}_{i_t|c}\hat{\mathbb{E}}_{k|c}\|\tilde{\mathbf{w}}_{c,t} - \mathbf{w}_{c,t}^k\|^2 \leq 4\eta_t^2 H^2 G^2.$$

*Proof.* Let $t_0 = t - (t \mod H)$ and recall that by synchronization we have $\mathbf{w}_{c,t_0}^k = \bar{\mathbf{w}}_{c,t_0} = \tilde{\mathbf{w}}_{c,t_0}$. Using $\mathbb{E}\|X - \mathbb{E}X\|^2 = \mathbb{E}\|X\|^2 - \|\mathbb{E}X\|^2$ with $X = \mathbf{w}_{c,t}^k - \mathbf{w}_{c,t_0}^k$, we get

$$\mathbb{E}_{i_t|c}\mathbb{E}_{k|c}\|\bar{\mathbf{w}}_{c,t} - \mathbf{w}_{c,t}^k\|^2 = \mathbb{E}_{i_t|c}\mathbb{E}_{k|c}\|\mathbf{w}_{c,t}^k - \mathbf{w}_{c,t_0}^k - (\bar{\mathbf{w}}_{c,t} - \mathbf{w}_{c,t_0}^k)\|^2$$

$$= \mathbb{E}_{i_t|c}\mathbb{E}_{k|c}\|\mathbf{w}_{c,t}^k - \mathbf{w}_{c,t_0}^k\|^2 - \|\bar{\mathbf{w}}_{c,t} - \bar{\mathbf{w}}_{c,t_0}\|^2$$

$$\leq \mathbb{E}_{i_t|c}\mathbb{E}_{k|c}\|\mathbf{w}_{c,t}^k - \mathbf{w}_{c,t_0}^k\|^2$$

$$= \mathbb{E}_{k|c}\mathbb{E}_{i_t|c}\|\sum_{\tau=t_0}^{t-1} \eta_\tau \nabla f^{i_\tau}(\mathbf{w}_{c,\tau}^k)\|^2$$

$$\leq 4\eta_t^2 H \sum_{\tau=t_0}^{t-1} \mathbb{E}_{k|c}\mathbb{E}_{i_t|c}\|\nabla f^{i_\tau}(\mathbf{w}_{c,\tau}^k)\|^2 \tag{18}$$

$$\overset{4.3}{\leq} 4\eta_t^2 H^2 G^2,$$

where (18) uses $\eta_t \leq \eta_{t_0} \leq 2\eta_t$ and $\|\sum_{i=1}^H \mathbf{a}_i\|^2 \leq H\sum_{i=1}^H \|\mathbf{a}_i\|^2$. Note that the bound for $\mathbb{E}_{i_t|c}\hat{\mathbb{E}}_{k|c}\|\tilde{\mathbf{w}}_{c,t} - \mathbf{w}_{c,t}^k\|^2$ follows using the same argument. The assumption about the learning rate implies that it does not decay by more than some factor (e.g., $\frac{1}{2}$) before the next synchronization, which can be easily satisfied by adding $H$ in the denominator of $\eta_t$. $\square$

**Lemma B.4** (Descent bound 2). *Assume that $\eta_{t+1} \leq \eta_t$ and $\eta_{t_0} \leq 2\eta_t$, where $t_0 = t - (t \mod H)$ and $\boldsymbol{\delta}_{c,t}$ defined as in Lemma B.2. Then,*

$$\|\tilde{\mathbf{w}}_{c,t+1} - \mathbf{w}_c^*\|^2 \leq (1 - \eta_t\mu)\|\tilde{\mathbf{w}}_{c,t} - \mathbf{w}_c^*\|^2 + 4\eta_t^2\sigma^2\|\mathbf{p}_c\|^2 + 4\eta_t^2 G^2\|\boldsymbol{\delta}_{c,t}\|_1^2 + 8L\eta_t^3 H^2 G^2$$

$$+ \eta_t \sum_{k=1}^{K} (4L\eta_t\mathbf{p}_c^k - \hat{\mathbf{p}}_{c,t}^k)[f_c(\mathbf{w}_{c,t}^k) - f_c(\mathbf{w}_c^*)].$$

*Proof.* The bound follows from applying Lemmas B.2 and B.3 on Lemma B.1. □

**Discussion** Let us stop here and compare this bound with that of vanilla local SGD. First, we observe that we retrieve the original variance reduction (up to a constant factor). Next, we see that the directly incurred cost from aggregation mismatch is $G^2$. The aggregation mismatch also manifests in the optimality gap in the sense that it "dampens" the guarantee as the aggregation increases, which we will make more precise later.

In general, the descent lemma above can recover local SGD's descent lemma in the (FL) setting, which immediately implies its convergence rate.

**Corollary B.5.** *Lemma B.4 recovers local SGD descent lemma from [36, Lemma 3.1] up to constant factors.*

*Proof.* Since $C = 1$ in local SGD, this trivially gives "uniform" routing and thus $\|\boldsymbol{\delta}_{c,t}\|_1 = 0$, i.e., $\hat{\mathbf{p}}_{c,t}^k = \mathbf{p}_c^k = 1/K$. Note that we have used the same assumptions, so we can apply Lemma B.4 with $\eta_t \leq \frac{1}{16L}$ to obtain the descent lemma of local SGD up to constant factors (and up to application of Lemmas B.2 and B.3). □

Next, we want to bound the quantity $\|\boldsymbol{\delta}_{c,t}\|_1^2$ given the update (4), and relate the new bound to Lemma B.4. After that, we can derive the convergence rate with the help of a technical lemma. We also derive the convergence rate given a slow decay assumption on $\|\boldsymbol{\delta}_{c,t}\|_1^2$, which shows more clearly the effect of aggregation mismatch on convergence.

## B.2. Bounding the total variation distance

The following bound follows from the router update in (4).

**Lemma B.6** (Total variation distance bound). *Consider the choice $\hat{\boldsymbol{\pi}}_{c,t}^k = \frac{\exp(-\eta_t f_c(\mathbf{w}_{c,t}^k))}{\sum_{c'=1}^{C}\exp(-\eta_t f_c(\mathbf{w}_{c',t}^k))}$, and assume that we can write $\boldsymbol{\pi}_c^k = \frac{\exp(-\bar{\eta} f_c(\mathbf{w}_c^*))}{\sum_{c'=1}^{C}\exp(-\bar{\eta} f_c(\mathbf{w}_{c'}^*))}$, where $f_c$ is bounded below by 0 and $\bar{\eta} \geq \eta_t$, $\forall t \geq 0$. Then,*

$$\|\boldsymbol{\delta}_{c,t}\|_1^2 \leq 4\eta_t\mathbb{E}_{k|c}[f_c(\mathbf{w}_{c,t}^k) - f_c(\mathbf{w}_c^*)] + \bar{\eta} f_c(\mathbf{w}_c^*) + 4\log KC + 10L\eta_t^3 H^2 G^2.$$

*Proof.* Let $t_0 = t - (t \mod H)$. We will consider the cases where $t = t_0$ and $t > t_0$ separately.

**Case $t = t_0$:** Let $\hat{Z}_t^k = \sum_{c'=1}^{C}\exp(-\eta_t f_c(\mathbf{w}_{c',t}^k))$ be the partition function of client $k$, so that we can write $\hat{\boldsymbol{\pi}}_{c,t}^k = \exp(-\eta_t f_c(\mathbf{w}_{c,t}^k))/\hat{Z}_t^k$. Recall that $\mathbf{p}_c^k = p(k|c) \propto N^k \boldsymbol{\pi}_c^k$, and let $\hat{Z}_{c,t} = \sum_{k=1}^{K} N^k \hat{\boldsymbol{\pi}}_{c,t}^k$ be the partition function of cluster $c$. Equivalently define $Z^k$ and $Z_{c,t}$ to be the partition functions of

client $k$ and cluster $c$ give the optimal router $\boldsymbol{\pi}_c^k$, respectively. Observe

$$\|\boldsymbol{\delta}_{c,t}\|_1^2 = \left(\sum_{k=1}^K |\hat{\mathbf{p}}_{c,t}^k - \mathbf{p}_c^k|\right)^2 = \left(\sum_{k=1}^K \mathbf{p}_c^k |\hat{\mathbf{p}}_{c,t}^k/\mathbf{p}_c^k - 1|\right)^2$$

$$\leq 2\sum_{k=1}^K \mathbf{p}_c^k \log\frac{\mathbf{p}_c^k}{\hat{\mathbf{p}}_{c,t}^k}$$

$$= 2\sum_{k=1}^K \mathbf{p}_c^k (\eta_t f_c(\mathbf{w}_{c,t}^k) - \bar{\eta} f_c(\mathbf{w}_c^*) + \log\frac{\hat{Z}_t^k}{Z^k} + \log\frac{\hat{Z}_{c,t}}{Z_c})$$

$$\leq 2\sum_{k=1}^K \mathbf{p}_c^k \eta_t (f_c(\mathbf{w}_{c,t}^k) - f_c(\mathbf{w}_c^*)) + \log\frac{\hat{Z}_t^k}{Z^k} + \log\frac{\hat{Z}_{c,t}}{Z_c}, \tag{19}$$

where we have used Pinsker's inequality, the router's expression, and $\bar{\eta} \geq \eta_t$.

Using $\max_{1\leq k\leq K}\{x_k\} \leq \log\sum_{k=1}^K \exp(x_k) \leq \max_{1\leq k\leq K}\{x_k\} + \log K$, we can write

$$\log\frac{\hat{Z}_t^k}{Z^k} = \log\sum_{c'=1}^C \exp(-\eta_t f_c(\mathbf{w}_{c',t}^k)) - \log\sum_{c'=1}^C \exp(-\bar{\eta} f_c(\mathbf{w}_{c'}^*))$$

$$\leq \max_{1\leq c'\leq C}\{-\eta_t f_c(\mathbf{w}_{c',t}^k)\} - \max_{1\leq c'\leq C}\{-\bar{\eta} f_c(\mathbf{w}_{c'}^*)\} + \log C$$

$$= \min_{1\leq c'\leq C}\{\bar{\eta} f_c(\mathbf{w}_{c'}^*)\} - \min_{1\leq c'\leq C}\{\eta_t f_c(\mathbf{w}_{c',t}^k)\} + \log C, \tag{$\dagger$}$$

and using similar arguments, we can show that

$$\log\frac{\hat{Z}_{c,t}}{Z_c} = \log\sum_{k=1}^K N^k \hat{\boldsymbol{\pi}}_{c,t}^k - \log\sum_{k=1}^K N^k \boldsymbol{\pi}_c^k$$

$$= \log\sum_{k=1}^K \exp(-\eta_t f_c(\mathbf{w}_{c,t}^k) + \log\frac{N^k}{\hat{Z}_t^k}) - \log\sum_{k=1}^K \exp(-\bar{\eta} f_c(\mathbf{w}_c^*) + \log\frac{N^k}{Z^k})$$

$$= \min_{1\leq k\leq K}\{\bar{\eta} f_c(\mathbf{w}_c^*) + \log\frac{Z^k}{N^k}\} - \min_{1\leq k\leq K}\{\eta_t f_c(\mathbf{w}_{c,t}^k) + \log\frac{\hat{Z}_t^k}{N^k}\} + \log K. \tag{$*$}$$

By properties of the LogSumExp function, we have

$$-\min_{1\leq c'\leq C}\{\bar{\eta} f_c(\mathbf{w}_{c'}^*)\} \leq \log Z^k \leq -\min_{1\leq c'\leq C}\{\bar{\eta} f_c(\mathbf{w}_{c'}^*)\} + \log C,$$

and similarly with $\log\hat{Z}_t^k$ and $-\eta_t\min_{1\leq c'\leq C} f_c(\mathbf{w}_{c',t}^k)$. Observe that $-\log\frac{N}{N^k} \leq 0$ and $\min_{1\leq k\leq K}\{\log\frac{N}{N^k}\} \leq \log K$ since the uniform case has the lowest max probability. Now define the centered function $f_c^\circ(\cdot) := f_c(\cdot) - \min_{1\leq c'\leq C}\{f_c(\cdot)\}$ and note that $f_c^\circ(\cdot) \leq f_c(\cdot)$. Adding and subtracting $\log N$ to both terms in $(*)$ and using the expressions above, we can get

$$\log\frac{\hat{Z}_{c,t}}{Z_c} \leq \min_{1\leq k\leq K}\{\bar{\eta} f_c(\mathbf{w}_c^*) + \log\frac{NZ^k}{N^k}\} - \min_{1\leq k\leq K}\{\eta_t f_c(\mathbf{w}_{c,t}^k) + \log\frac{N\hat{Z}_t^k}{N^k}\} + \log K$$

$$\leq \bar{\eta} f_c^\circ(\mathbf{w}_c^*) - \min_{1\leq k\leq K}\{\eta_t f_c^\circ(\mathbf{w}_{c,t}^k)\} + 2\log K + \log C. \tag{$\dagger\dagger$}$$

Combining $(\dagger)$ and $(\dagger\dagger)$, we get

$$\log\frac{\hat{Z}_t^k}{Z^k} + \log\frac{\hat{Z}_{c,t}}{Z_c} \leq \bar{\eta} f_c(\mathbf{w}_c^*) - \min_{1\leq c'\leq C}\{\eta_t f_c(\mathbf{w}_{c',t}^k)\} - \min_{1\leq k\leq K}\{\eta_t f_c(\mathbf{w}_{c,t}^k)\} + 2\log KC$$

$$\leq \bar{\eta} f_c(\mathbf{w}_c^*) + 2\log KC.$$

where the second inequality follows because $\min_i\{A_i + B_i\} \leq \min_i\{A_i\} + \min_i\{B_i\}$. Applying this inequality to the overall bound (19), we have

$$\|\boldsymbol{\delta}_{c,t}\|_1^2 \leq 2\eta_t \mathbb{E}_{k|c}[f_c(\mathbf{w}_{c,t}^k) - f_c(\mathbf{w}_c^*)] + 2\bar{\eta} f_c(\mathbf{w}_c^*) + 4\log KC. \tag{20}$$

**Case** $t > t_0$: Note that $\boldsymbol{\delta}_{c,t} = \boldsymbol{\delta}_{c,t_0}$ by (4), so we get the same bound (20) but in terms of $t_0$. If we decompose the function gap $\mathbb{E}_{k|c}[f_c(\mathbf{w}_{c,t_0}^k) - f_c(\mathbf{w}_c^*)] = \mathbb{E}_{k|c}[f_c(\mathbf{w}_{c,t_0}^k) - \mathbb{E}_{i_t|c}f_c(\mathbf{w}_{c,t}^k)] + \mathbb{E}_{k|c}[\mathbb{E}_{i_t|c}f_c(\mathbf{w}_{c,t}^k) - f_c(\mathbf{w}_c^*)]$, we see that it suffices to bound the first term to be able to write $\boldsymbol{\delta}_{c,t}$ in terms of function gap at step $t$. We can also take the expectations $\mathbb{E}_{i_t|c}$ out since neither $\mathbf{w}_{c,t_0}^k$ nor $\mathbf{w}_c^*$ depend on $i_t = \cdots = i_{t_0}$.

Recall that $\mathbf{w}_{c,t_0}^k = \mathbb{E}_{k|c}[\mathbf{w}_{c,t_0}^k]$. Using $L$-smoothness from Assumption 4.2, we get

$$
\mathbb{E}_{k,i_t|c}[f_c(\mathbf{w}_{c,t_0}^k) - f_c(\mathbf{w}_{c,t}^k)] \leq \mathbb{E}_{k,i_t|c}\langle \nabla f_c(\mathbf{w}_{c,t}^k), \mathbf{w}_{c,t_0}^k - \mathbf{w}_{c,t}^k\rangle + \frac{L}{2}\mathbb{E}_{k,i_t|c}\|\mathbf{w}_{c,t_0}^k - \mathbf{w}_{c,t}^k\|^2
$$

$$
\overset{\text{(Young)}}{\leq} \gamma^{-1}\mathbb{E}_{k|c}\|\nabla f_c(\mathbf{w}_{c,t}^k)\|^2 + (\gamma + \frac{L}{2})\mathbb{E}_{k,i_t|c}\|\mathbf{w}_{c,t_0}^k - \mathbf{w}_{c,t}^k\|^2
$$

$$
\overset{(18)}{\leq} \gamma^{-1}\mathbb{E}_{k|c}\|\nabla f_c(\mathbf{w}_{c,t}^k)\|^2 + (\gamma + \frac{L}{2})4\eta_t^2 H^2 G^2
$$

$$
\overset{(12)}{\leq} \mathbb{E}_{k|c}[f_c(\mathbf{w}_{c,t}^k) - f_c(\mathbf{w}_c^*)] + 10L\eta_t^2 H^2 G^2,
$$

where we have chosen $\gamma := 2L$.

We complete the proof by taking the max of both cases, which simply amounts to adding both cases. $\qquad\square$

The following descent lemma will be used to get the convergence rate without any assumptions on $\|\boldsymbol{\delta}_{c,t}\|_1$ other than what we have in the router update (4).

**Lemma B.7** (Descent bound 3)**.** *Let the conditions in Lemmas B.4 and B.6 be satisfied. Without loss of generality, assume that $f_c(\mathbf{w}_c^*) = 0$. If $\eta_t \leq \frac{\gamma_t}{\max\{\frac{5}{2}, 16G^2, 4L\}}$, where $\gamma_t = \min\{1, \min_{k \in [K];\ \mathbf{p}_c^k > 0}\{\hat{\mathbf{p}}_{c,t}^k/\mathbf{p}_c^k\}\}$, then*

$$
\mathbb{E}_{i_t|c}\|\tilde{\mathbf{w}}_{c,t+1} - \mathbf{w}_c^*\|^2 \leq (1 - \eta_t\mu)\|\tilde{\mathbf{w}}_{c,t} - \mathbf{w}_c^*\|^2 - \frac{1}{2}\eta_t\hat{\mathbb{E}}_{k|c}[f_c(\mathbf{w}_{c,t}^k) - f_c(\mathbf{w}_c^*)]
$$
$$
+ 4\eta_t^2\sigma^2\|\mathbf{p}_c\|^2 + 16\eta_t^2 G^2 \log KC + 9\eta_t^3 LH^2 G^2.
$$

*Proof.* We apply Lemma B.6 on Lemma B.4 and rearrange to get

$$
\mathbb{E}_{i_t|c}\|\tilde{\mathbf{w}}_{c,t+1} - \mathbf{w}_c^*\|^2 \leq (1 - \eta_t\mu)\|\tilde{\mathbf{w}}_{c,t} - \mathbf{w}_c^*\|^2 + 4\eta_t^2\sigma^2\|\mathbf{p}_c\|^2 + 16\eta_t^2 G^2 \log KC
$$
$$
+ 40L\eta_t^5 H^2 G^4 + 8L\eta_t^3 H^2 G^2
$$
$$
+ \eta_t\sum_{k=1}^{K}(16\eta_t^2 G^2\mathbf{p}_c^k + 4L\eta_t\mathbf{p}_c^k - \hat{\mathbf{p}}_{c,t}^k)[f_c(\mathbf{w}_{c,t}^k) - f_c(\mathbf{w}_c^*)].
$$

In order to have a meaningful convergence of the optimality gap, we have to bound it from above, so we should have

$$
\mathbf{p}_c^k(16\eta_t^2 G^2 + 8L\eta_t - 1) + \mathbf{p}_c^k - \hat{\mathbf{p}}_{c,t}^k < -A, \tag{21}
$$

for some $A > 0$.

Suppose $\mathbf{p}_c^k > \hat{\mathbf{p}}_{c,t}^k$, and recall that $c$ is given, so we fix it. Write $r_t^k := \hat{\mathbf{p}}_{c,t}^k/\mathbf{p}_c^k < 1$. Then, $\mathbf{p}_c^k - \hat{\mathbf{p}}_{c,t}^k = (1 - r_t^k)\mathbf{p}_c^k < \mathbf{p}_c^k$, so that (21) becomes $\mathbf{p}_c^k(16\eta_t^2 G^2 + 4L\eta_t - r_t^k) < -A$. If we set $\eta_t \leq \frac{B}{\max\{16G^2, 4L\}}$ for some $B > 0$, we would have $\mathbf{p}_c^k(B\eta_t + B - r_t^k) < -A$, implying that

$$
\eta_t < \frac{1}{B}(r_t^k - B - A/\mathbf{p}_c^k) = \frac{\hat{\mathbf{p}}_{c,t}^k - A}{B\mathbf{p}_c^k} - 1.
$$

Setting $A = (1 - (\bar{\eta} + 1)(r_t^k)^{-1}B)\hat{\mathbf{p}}_{c,t}^k > 0$ gives $\eta_t < \bar{\eta}$, where $\bar{\eta}$ is some strict upper bound of $\eta_t$ for all $t$, but we should also have $(\bar{\eta} + 1)(r_t^k)^{-1}B < 1 \iff B < \frac{r_t^k}{\bar{\eta}+1}$. Thus, we set $B := \frac{r_t^k}{\bar{\eta}+2}$, getting

$$
A = \frac{1}{2}\hat{\mathbf{p}}_{c,t}^k, \qquad \eta_t \leq \frac{r_t^k}{(\bar{\eta} + 2)\max\{16G^2, 4L\}}. \tag{22}
$$

Now, if $\mathbf{p}_c^k \leq \hat{\mathbf{p}}_{c,t}^k$ with $\eta_t \leq \frac{D}{\max\{16G^2, 4L\}}$ for some $D > 0$, (21) would imply $\eta_t < \frac{1-A}{D\mathbf{p}_c^k} - 1$, so that $A = (1 - (\bar{\eta} + 1)D)\mathbf{p}_c^k > 0$ gives $\eta_t < \bar{\eta}$ but under the condition $D < \frac{1}{\bar{\eta}+1}$. Thus, setting $D := \frac{1}{\bar{\eta}+2}$ gives the same setting in (22) with 1 instead of $r_t^k$. Thus, for all $k \in [K]$, we should have

$$\eta_t \leq \frac{\min\{1, r_t^k\}}{(\bar{\eta} + 2)\max\{16G^2, 4L\}}.$$

We can restrict the denominator to $\max\{1, 16G^2, 4L\}$ without loss of generality. Then, we can upper bound $\eta_t \leq \frac{1}{(\bar{\eta}+2)} < \frac{1}{2}$, so that $\bar{\eta} = \frac{1}{2}$ suffices for this choice.

Overall, we have $\eta_t \leq \frac{\min\{1, \min_{k \in [K]}\{\hat{\mathbf{p}}_{c,t}^k / \mathbf{p}_c^k\}\}}{\max\{\frac{5}{2}, 16G^2, 4L\}}$, and using (21) with $A = \frac{1}{2}\hat{\mathbf{p}}_{c,t}^k$, we get

$$\mathbb{E}_{i_t|c}\|\tilde{\mathbf{w}}_{c,t+1} - \mathbf{w}_c^*\|^2 \leq (1 - \eta_t\mu)\|\tilde{\mathbf{w}}_{c,t} - \mathbf{w}_c^*\|^2 - \frac{1}{2}\eta_t\hat{\mathbb{E}}_{k|c}[f_c(\mathbf{w}_{c,t}^k) - f_c(\mathbf{w}_c^*)]$$
$$+ 4\eta_t^2\sigma^2\|\mathbf{p}_c\|^2 + 16\eta_t^2 G^2 \log KC + 8L\eta_t^3 H^2 G^2 + 40L\eta_t^5 H^2 G^4.$$

Given our choice of $\eta_t$, we note that $40L\eta_t^5 H^2 G^4 \leq L\eta_t^4 H^2 G^2 \leq L\eta_t^3 H^2 G^2$, which completes the proof. $\qquad\square$

**Discussion** Note that the learning rate is not as strict as it may look. First of all, note that $\hat{\mathbf{p}}_{c,t}^k < \mathbf{p}_c^k$ is the case of interest, as otherwise, $\gamma_t = 1$. Taking the minimum for $k$ such that $\mathbf{p}_c^k > 0$ makes sense because $\hat{\mathbf{p}}_{c,t}^k \geq \mathbf{p}_c^k = 0$, so $\gamma_t = 1$.

Now assume that $\hat{\mathbf{p}}_{c,t}^k \leq \mathbf{p}_c^k$ for all $t$. The lowest value $\gamma_t$ can attain is when $\mathbf{p}_c^k = 1$ and $\hat{\mathbf{p}}_{c,t}^k$ is very small. This can happen, for example, when a cluster has one client. However, a uniform initialization for the routers would have that

$$\hat{\boldsymbol{\pi}}_{c,0}^k = 1/C \quad \implies \quad \hat{\mathbf{p}}_{c,0}^k = \frac{p(k)\hat{\boldsymbol{\pi}}_{c,0}^k}{\sum_{k'=1}^K p(k')\hat{\boldsymbol{\pi}}_{c,0}^{k'}} = p(k).$$

Since $\mathbf{p}_c^k = \frac{p(k)\boldsymbol{\pi}_c^k}{p(c)}$, we would then have $\hat{\mathbf{p}}_{c,0}^k / \mathbf{p}_c^k = \frac{p(c)}{\boldsymbol{\pi}_c^k} \leq 1$ since we assumed $\hat{\mathbf{p}}_{c,t}^k \leq \mathbf{p}_c^k$.

Suppose $\boldsymbol{\pi}_c^k = 1$. If $\sum_{k=1}^K \boldsymbol{\pi}_c^k = 1$, i.e., the number of clients in cluster $c$ is 1, then we cannot improve $\hat{\mathbf{p}}_{c,0}^k / \mathbf{p}_c^k = p(k)$ any further. This can be even worse if there is one data point for client $k$. However, these extreme heterogeneity scenarios are inherently difficult, so it is better to capture this heterogeneity with some term, particularly when $\boldsymbol{\pi}_c^k \geq p(c)$, which follows from $\hat{\mathbf{p}}_{c,t}^k \leq \mathbf{p}_c^k$.

For example, assume that $\boldsymbol{\pi}_c^k \leq U_c^{-1}p(c)$ for all $k$ such that $\boldsymbol{\pi}_c^k \geq p(c)$, so that $U_c \in [p(c), 1]$. In other words, we can choose $U_c = \min_{k; p(c) \leq \boldsymbol{\pi}_c^k}\{p(c)/\boldsymbol{\pi}_c^k\}$. This implies that $\hat{\mathbf{p}}_{c,0}^k / \mathbf{p}_c^k = \frac{p(c)}{\boldsymbol{\pi}_c^k} \geq U_c$.

The value $U_c$ is a uniformity measure, so that a larger $U_c$ denotes a more uniform allocation of clients in cluster $c$. For example, if $U_c = 1$, then, for all $k$ such that $\boldsymbol{\pi}_c^k \geq p(c)$, we have $\boldsymbol{\pi}_c^k = p(c)$. On the other hand, if $U_c = p(c)$, then it is possible for some clients $k$ to have $\boldsymbol{\pi}_c^k = 1$, or in the worst case, $p(c) = p(k)$ when only one client is in cluster $c$ (remember that clients with $\boldsymbol{\pi}_c^k < p(c)$ are ignored). When cluster sizes are comparable, we have $p(c) \approx 1/C$, meaning that $U_c \geq 1/C$.

Thus, in general, with uniform router initialization and when $|\mathbf{p}_c^k - \hat{\mathbf{p}}_{c,t}^k| \leq |\mathbf{p}_c^k - \hat{\mathbf{p}}_{c,0}^k|$ (which is a mild restriction to ensure $\hat{\mathbf{p}}_{c,0}^k / \mathbf{p}_c^k$ is smaller than $\hat{\mathbf{p}}_{c,0}^k / \mathbf{p}_c^k$), we have

$$U_c = \min_{k; p(c) \leq \boldsymbol{\pi}_c^k}\{p(c)/\boldsymbol{\pi}_c^k\} \leq \min\{1, \min_{k \in [K]}\{p(k)/\mathbf{p}_c^k\}\} \leq \gamma_t \leq 1. \tag{23}$$

Regarding the $\min$ operator in $U_c$, it is only required because it is a uniform learning rate for all clients, so it must converge for the worst client, which is the client with the least amount of data (i.e., lowest $p(k)$). Thus, we believe that this can be removed when considering learning rates per client. We leave this analysis for future work.

## B.3. Convergence rates

In order to get convergence rates from descent lemmas, we make use of the following useful lemma, which is based on [44, Lemma 3.3].

**Lemma B.8.** *Let $\{a_t\}_{t\geq 0}$, $\{b_t\}_{t\geq 0}$, and $\{c_t\}_{t\geq 0}$, be arbitrary non-negative sequences such that*

$$a_{t+1} \leq (1 - \mu\eta_t)a_t - \eta_t b_t + \eta_t^2 c_t.$$

*Let $\eta_t = \frac{\alpha}{t+s}$ for $t \geq 0$ and $s \geq 1$, and choose $\alpha = \frac{1}{\mu}$. Then, we have the following inequality*

$$\sum_{t=0}^{T-1} b_t \leq (s-1)\mu a_0 + \sum_{t=0}^{T-1} \eta_t c_t.$$

*Proof.* Let $r_t := 1 - \mu\eta_t$. Then,

$$\begin{aligned}
a_{t+1} &\leq r_t a_t - \eta_t b_t + \eta_t^2 c_t \\
&\leq r_t r_{t-1} a_{t-1} - \eta_t b_t - r_t \eta_{t-1} b_{t-1} + \eta_t^2 c_t + r_t \eta_{t-1}^2 c_{t-1} \\
&\leq -\eta_t b_t - r_t \eta_{t-1} b_{t-1} - r_t r_{t-1}\eta_{t-2} b_{t-2} \\
&\quad + \eta_t^2 c_t + r_t \eta_{t-1}^2 c_{t-1} + r_t r_{t-1}\eta_{t-2}^2 c_{t-2} + r_t r_{t-1} r_{t-2} a_{t-2} \\
&\leq \cdots \\
\implies a_{T+1} &\leq r_{0:T} a_0 + \sum_{t=0}^{T} r_{t+1:T}\eta_t(-b_t + \eta_t c_t),
\end{aligned}$$

where we denote $r_{t_1:t_2} := \prod_{t=t_1}^{t_2} r_t$, which defaults to 1 when $t_1 > t_2$.

Observe that

$$\sum_{t=t_1}^{t_2} \eta_t = \sum_{t=t_1}^{t_2} \frac{\alpha}{t+s} \geq \alpha \int_{t=t_1}^{t_2} \frac{1}{t+s} = \alpha \log\frac{t_2+s}{t_1+s}.$$

Hence,

$$r_{t_1:t_2} = \prod_{t=t_1}^{t_2}(1 - \mu\eta_t) \leq \prod_{t=t_1}^{t_2} \exp(-\mu\eta_t) = \exp(-\mu\sum_{t=t_1}^{t_2}\eta_t) \leq \left(\frac{t_1+s}{t_2+s}\right)^{\mu\alpha}.$$

We can confirm that for $\alpha = 1/\mu$, we have

$$r_t^{-1}\eta_t = \left(\frac{t+s}{t+s-\mu\alpha}\right)\left(\frac{\alpha}{t+s}\right) = \frac{\alpha}{t+s-1} = \eta_{t-1},$$

so that $r_t = \frac{\eta_t}{\eta_{t-1}}$. This implies $r_{t_1:t_2} = \frac{\eta_{t_2}}{\eta_{t_1-1}} = \frac{t_1+s-1}{t_2+s} \leq \frac{t_1+s}{t_2+s}$, so the inequality above is almost tight when $\alpha = 1/\mu$ (loose by a multiplicative factor of $\frac{t_1+s}{t_1+s-1}$). This also implies that $\eta_T = \eta_{T-1}r_T = \cdots = \eta_t r_{t+1:T} = \cdots = \eta_0 r_{1:T}$. so we can factor these terms out and divide both sides by $\eta_T$. Hence, we have $\frac{r_{0:T}}{\eta_T} = \frac{r_0}{\eta_0} = (s-1)\mu$, and by observing that $0 \leq \frac{a_{T+1}}{\eta_T}$, we can get the desired bound. $\qquad\square$

Now we are ready to prove the main theoretical results of the paper.

**Theorem B.9** (Convergence rate). *Consider the setup in Section 4. Let $\tilde{\sigma}^2 = \sigma^2\|\mathbf{p}_c\|^2$, $\kappa = \frac{L}{\mu}$, and $U_c = \min_{k;\, p(c)\leq\pi_c^k}\{p(c)/\pi_c^k\}$. Initialize $\hat{\pi}_{c,0}^k = 1/C$ for all $k \in [K]$, and assume $|\mathbf{p}_c^k - \hat{\mathbf{p}}_{c,t}^k| \leq |\mathbf{p}_c^k - \hat{\mathbf{p}}_{c,0}^k|$ for all $t \geq 0$. Assume that $f_c(\mathbf{w}_c^*) = 0$, without loss of generality. Let $\eta_t \leq \frac{\alpha}{t+s}$ with $\alpha = \frac{1}{\mu}$ and $s \geq \max\{3H, 4\kappa/U_c, 16G^2/\mu U_c\}$. Consider the weighted average after $T$ iterations $\hat{\mathbf{w}}_{c,T} := \frac{1}{\sum_{t=0}^{T-1} w_t} \sum_{t=0}^{T-1} w_t \tilde{\mathbf{w}}_{c,t}$ with $w_t = (t+s)^2$. Then, the following holds*

$$\mathbb{E}f_c(\hat{\mathbf{w}}_{c,T}) - f_c(\mathbf{w}_c^*) \leq (8\tilde{\sigma}^2 + 32G^2\log KC)(\frac{1}{\mu T} + \frac{2s-1}{\mu T^2})$$

$$+ \frac{18\kappa H^2 G^2}{\mu T^2} + \frac{24(s-1)s^2 G^2}{\mu T^3}.$$

*If $L \geq 4G^2$, we have the following asymptotic bound*

$$\mathbb{E}f_c(\hat{\mathbf{w}}_{c,T}) - f_c(\mathbf{w}_c^*) \leq \mathcal{O}\left(\frac{1}{\mu T} + \frac{\kappa/U_c + H}{\mu T^2}\right)\tilde{\sigma}^2 + \mathcal{O}\left(\frac{\log KC}{\mu T} + \frac{\kappa H^2}{\mu T^2} + \frac{(\kappa/U_c)^3 + H^3}{\mu T^3}\right)G^2.$$

*Proof.* Note that $\eta_t$ satisfies Lemma B.7 by construction of $s$ and (23). It also satisfies Lemma B.3 since for $t \in [t_0, t_0 + H)$, we have $\frac{\eta_t}{\eta_{t+H}} = \frac{t+s+H}{t+s} \leq 2$ because $s + t \geq s \geq H$.

Let $\{w_t\}_{t \geq 0}$ be a non-negative (averaging) sequence. We use Lemma B.8 on Lemma B.7 with

$$a_t = w_t \mathbb{E}_{i_t|c}\|\tilde{\mathbf{w}}_{c,t} - \mathbf{w}_c^*\|^2, \qquad b_t = \frac{w_t}{2}f_c(\tilde{\mathbf{w}}_{c,t}) - f_c(\mathbf{w}_c^*), \qquad c_t = w_t(A + B\eta_t),$$

where $A = 4\sigma^2\|\mathbf{p}_c\|^2 + 16G^2 \log KC$ and $B = 9LH^2G^2$. Note $f_c(\tilde{\mathbf{w}}_{c,t}) - f_c(\mathbf{w}_c^*) \leq \hat{\mathbb{E}}_{k|c}[f_c(\mathbf{w}_{c,t}^k) - f_c(\mathbf{w}_c^*)]$ by Jensen's inequality. Thus,

$$\sum_{t=0}^{T-1} w_t \mathbb{E}_{i_t|c}[f_c(\bar{\mathbf{w}}_{c,t}) - f_c(\mathbf{w}_c^*)] \leq 2(s-1)\mu w_0 a_0 + 2A\sum_{t=0}^{T-1} w_t \eta_t + 2B\sum_{t=0}^{T-1} w_t \eta_t^2.$$

From the expression above, it makes sense to choose $w_t = (t+s)^2$. Indeed,

$$\sum_{t=0}^{T-1} w_t \eta_t = \sum_{t=0}^{T-1} \alpha(t+s) = \frac{T(T-1)}{2\mu} + \frac{Ts}{\mu}, \quad \text{and} \quad \sum_{t=0}^{T-1} w_t \eta_t^2 = \sum_{t=0}^{T-1} \alpha^2 = \frac{T}{\mu^2}.$$

Hence, using Jensen's inequality with $\hat{\mathbf{w}}_{c,T} := \frac{1}{\sum_{t=0}^{T-1} w_t}\sum_{t=0}^{T-1} w_t \bar{\mathbf{w}}_{c,t}$ and letting $D = \|\tilde{\mathbf{w}}_{c,0} - \mathbf{w}_c^*\|$, we have with the tower property of conditional expectations that

$$\mathbb{E}f_c(\hat{\mathbf{w}}_{c,T}) - f_c(\mathbf{w}_c^*) \leq \frac{2(s-1)s^2\mu D^2}{\sum_{t=0}^{T-1} w_t} + 2A\left(\frac{T(T-1) + 2Ts}{2\mu \sum_{t=0}^{T-1} w_t}\right) + 2B\frac{T}{\mu^2 \sum_{t=0}^{T-1} w_t}.$$

We bound $\frac{1}{\sum_{t=0}^{T-1} w_t}$ using the fact $\sum_{t=0}^{T-1} w_t = \frac{1}{3}T^3 + (s - \frac{1}{2})T^2 + (s^2 - s + \frac{1}{6})T \geq \frac{1}{3}T^3$. Using this bound and plugging in $A$ and $B$, we get

$$\mathbb{E}f_c(\hat{\mathbf{w}}_{c,T}) - f_c(\mathbf{w}_c^*) \leq \frac{6(s-1)s^2\mu D^2}{T^3} + (8\sigma^2\|\mathbf{p}_c\|^2 + 32G^2 \log KC)\left(\frac{1}{\mu T} + \frac{2s-1}{\mu T^2}\right)$$

$$+ \frac{18LH^2G^2}{\mu^2 T^2}.$$

We use $\mu\mathbb{E}\|\tilde{\mathbf{w}}_{c,0} - \mathbf{w}_c^*\| \leq 2G$ [45, Lemma 2] and tower property of conditional expectation in terms of $\mathbb{E}_{i_t|c}$ to get the desired bound. $\square$

**Discussion** Note that in Theorem B.9, we have $\eta_t$ depending on $G^2$ and the bound has an extra $\mathcal{O}(\frac{G^2 \log KC}{\mu T})$ term in the asymptotic bound, which comes from Lemma B.6, where we bounded $\|\boldsymbol{\delta}_{c,t}\|_1^2$ using (4). Furthermore, the terms $U_c$ appear in our analysis, but we explain that they do not affect the recovery of local SGD rates. Indeed, in the (FL) case, $U_c \geq p(c) = 1$ since $C = 1$. Even if we have $C$ copies of (FL) with $p(c) = 1/C$, since $p(k|c) = p(k)$, we would still have $U_c = 1$ (see the definition in (23)). In the (CFL) case, if we have similar cluster sizes and client sizes, then $U_c = 1/C$, which is the (linear) price to pay for learning the clusters given the uniform router initialization. This dependence can be reduced further by taking into the decay of $\hat{\mathbf{p}}_{c,t}^k/\mathbf{p}_c^k$ instead of assuming uniform router initialization and non-increasing $\hat{\mathbf{p}}_{c,t}^k/\mathbf{p}_c^k$ in $t$, but we leave such an analysis for future work.

We now prove a stronger convergence rate given a stronger assumption on the decrease of $\|\boldsymbol{\delta}_{c,t}\|_1^2$. Namely, we assume that $\|\boldsymbol{\delta}_{c,t}\|_1^2 \leq (t+s)^{-\beta}\|\boldsymbol{\delta}_{c,0}\|_1^2$ for $\beta \in (0,1)$. This convergence rate does not require a dependence on $G^2$ in the learning rate, and it weakens $\mathcal{O}(\frac{G^2 \log KC}{\mu T})$ proportionally to $\beta$. This particular range of the exponent of $\beta$ maintains the extra term in the asymptotic rate with an explicit dependence on $\beta$. The exponent is bounded above by 1 for technical convenience, and we believe this condition can be easily removed. In any case, exponents of 1 or larger would make the extra terms incurred from $\|\boldsymbol{\delta}_{c,t}\|_1^2$ disappear asymptotically. Indeed, the original rate of local SGD can be exactly recovered when $\|\boldsymbol{\delta}_{c,t}\|_1^2$ decays quickly (where $U_c = 1$ as explained above). We now state the stronger convergence rate.

**Theorem B.10** (Convergence Rate with decreasing $\|\boldsymbol{\delta}_{c,t}\|_1^2$). *Consider the setup in Section 4. Let $\tilde{\sigma}^2 = \sigma^2 \|\mathbf{p}_c\|^2$, $\kappa = \frac{L}{\mu}$, and $U_c = \min_{k;\,p(c)\leq\boldsymbol{\pi}_c^k}\{p(c)/\boldsymbol{\pi}_c^k\}$. Initialize $\hat{\boldsymbol{\pi}}_{c,0}^k = 1/C$ for all $k \in [K]$, and assume $|\mathbf{p}_c^k - \hat{\mathbf{p}}_{c,t}^k| \leq |\mathbf{p}_c^k - \hat{\mathbf{p}}_{c,0}^k|$ for all $t \geq 0$. Assume that $f_c(\mathbf{w}_c^*) = 0$ without loss of generality, and assume that $\|\boldsymbol{\delta}_{c,t}\|_1^2 \leq (t+s)^{-\beta}\|\boldsymbol{\delta}_{c,0}\|_1^2$ for $\beta \in (0,1)$. Let $\eta_t \leq \frac{\alpha}{t+s}$ with $\alpha = \frac{1}{\mu}$ and $s \geq \max\{3H, 4\kappa/U_c\}$. Consider the weighted average after $T$ iterations $\hat{\mathbf{w}}_{c,T} := \frac{1}{\sum_{t=0}^{T-1} w_t}\sum_{t=0}^{T-1} w_t \tilde{\mathbf{w}}_{c,t}$ with $w_t = (t+s)^2$. Then,*

$$\mathbb{E}f_c(\hat{\mathbf{w}}_{c,T}) - f_c(\mathbf{w}_c^*) \leq 12\tilde{\sigma}^2\left(\frac{1}{\mu T} + \frac{2s-1}{\mu T^2}\right) + 48G^2\frac{\|\boldsymbol{\delta}_{c,0}\|_1^2}{\mu T^{1+\beta}}$$
$$+ 24G^2\frac{(s-1+2\|\boldsymbol{\delta}_{c,0}\|_1^2 s^{-\beta})s^2}{\mu T^3} + 48LH^2G^2\frac{1}{\mu^2 T^2}.$$

*Asymptotically,*

$$\mathbb{E}f_c(\hat{\mathbf{w}}_{c,T}) - f_c(\mathbf{w}_c^*) \leq \mathcal{O}\left(\frac{1}{\mu T} + \frac{\kappa/U_c + H}{\mu T^2}\right)\tilde{\sigma}^2 + \mathcal{O}\left(\frac{\kappa H^2}{\mu T^2} + \frac{(\kappa/U_c)^3 + H^3}{\mu T^3}\right)G^2$$
$$+ \mathcal{O}\left(\frac{1}{\mu T^{1+\beta}} + \frac{(\kappa/U_c)^{2-\beta} + H^{2-\beta}}{\mu T^3}\right)\|\boldsymbol{\delta}_{c,0}\|_1^2 G^2.$$

*Proof.* Recall Lemma B.4

$$\|\tilde{\mathbf{w}}_{c,t+1} - \mathbf{w}_c^*\|^2 \leq (1 - \eta_t\mu)\|\tilde{\mathbf{w}}_{c,t} - \mathbf{w}_c^*\|^2 + 4\eta_t^2\sigma^2\|\mathbf{p}_c\|^2 + 4\eta_t^2 G^2\|\boldsymbol{\delta}_{c,t}\|_1^2 + 8L\eta_t^3 H^2 G^2$$
$$+ \eta_t\sum_{k=1}^K(4L\eta_t\mathbf{p}_c^k - \hat{\mathbf{p}}_{c,t}^k)[f_c(\mathbf{w}_{c,t}^k) - f_c(\mathbf{w}_c^*)].$$

We use the exact same reasoning in Lemma B.7 to get that $\eta_t \leq \frac{\min\{1,\min_{k\in[K]}\{\hat{\mathbf{p}}_{c,t}^k/\mathbf{p}_c^k\}\}}{\max\{5/2,4L\}}$. Our choice of $\eta_t$ already satisfies this rate from (23), and it clearly satisfies Lemma B.3 by construction of $s$. Thus, the overall bound becomes

$$\frac{1}{2}f_c(\tilde{\mathbf{w}}_{c,t}) - f_c(\mathbf{w}_c^*) \leq \frac{1}{2}\eta_t\hat{\mathbb{E}}_{k|c}[f_c(\mathbf{w}_{c,t}^k) - f_c(\mathbf{w}_c^*)]$$
$$\leq (1 - \eta_t\mu)\|\tilde{\mathbf{w}}_{c,t} - \mathbf{w}_c^*\|^2 + 4\eta_t^2\sigma^2\|\mathbf{p}_c\|^2 + 4\eta_t^2 G^2\|\boldsymbol{\delta}_{c,t}\|_1^2 + 8L\eta_t^3 H^2 G^2.$$

We can now invoke Lemma B.8 with

$$a_t = w_t\mathbb{E}_{i_t|c}\|\tilde{\mathbf{w}}_{c,t} - \mathbf{w}_c^*\|^2, \qquad b_t = \frac{w_t}{2}\mathbb{E}_{i_t|c}[f_c(\bar{\mathbf{w}}_{c,t}) - f_c(\mathbf{w}_c^*)], \qquad c_t = w_t(A_t + B\eta_t),$$

where $\{w_t\}_{t\geq 0}$ is an averaging sequence, $A_t = 4\sigma^2\|\mathbf{p}_c\|^2 + 4G^2\|\boldsymbol{\delta}_{c,t}\|_1^2$, and $B = 8LH^2G^2$. Thus,

$$\sum_{t=0}^{T-1} w_t\mathbb{E}_{i_t|c}[f_c(\bar{\mathbf{w}}_{c,t}) - f_c(\mathbf{w}_c^*)] \leq 2(s-1)\mu w_0 a_0 + 2\sum_{t=0}^{T-1} w_t\eta_t A_t + 2\sum_{t=0}^{T-1} w_t\eta_t^2 B.$$

We choose $w_t = (t+s)^2$ as in Theorem B.9 and use the assumption that $\|\boldsymbol{\delta}_{c,t}\|_1^2 \leq (t+s)^{-\beta}\|\boldsymbol{\delta}_{c,0}\|_1^2$ for $\beta \in (0,1)$ to get

$$\sum_{t=0}^{T-1} w_t\eta_t A_t = \alpha\sum_{t=0}^{T-1}(t+s)A_t = 4\alpha\sigma^2\|\mathbf{p}_c\|^2\sum_{t=0}^{T-1}(t+s) + 4\alpha G^2\sum_{t=0}^{T-1}(t+s)\|\boldsymbol{\delta}_{c,t}\|_1^2$$
$$= 4\alpha\sigma^2\|\mathbf{p}_c\|^2\left(\frac{T(T-1)}{2} + Ts\right) + 4\alpha G^2\|\boldsymbol{\delta}_{c,0}\|_1^2\sum_{t=0}^{T-1}(t+s)^{1-\beta}.$$

Furthermore,

$$\sum_{t=0}^{T-1}(t+s)^{1-\beta} \leq \int_0^T(t+s)^{1-\beta}dt = \frac{1}{2-\beta}((T+s)^{2-\beta} - s^{2-\beta}) \leq (T+s)^{2-\beta}.$$

Hence,

$$\sum_{t=0}^{T-1} w_t \eta_t A_t \leq 2\alpha\sigma^2 \|\mathbf{p}_c\|^2 T(T+2s-1) + 8\alpha G^2 \|\boldsymbol{\delta}_{c,0}\|_1^2 (T^{2-\beta} + s^{2-\beta}),$$

where we have used $(T+s)^{2-\beta} \leq 2\max\{T,s\}^{2-\beta} \leq 2(T^{2-\beta} + s^{2-\beta})$.

On the other hand, using $\sum_{t=0}^{T-1} \frac{1}{(t+s)^\beta} \leq T$, we get

$$\sum_{t=0}^{T-1} w_t \eta_t^2 B \leq 8\alpha^2 LH^2 G^2 T.$$

Using the averaging $\hat{\mathbf{w}}_{c,T} := \frac{1}{\sum_{t=0}^{T-1} w_t} \sum_{t=0}^{T-1} w_t \bar{\mathbf{w}}_{c,t}$, the fact that $\sum_{t=0}^{T-1} w_t \geq \frac{1}{3} T^3$, and $\mu\mathbb{E}\|\tilde{\mathbf{w}}_{c,0} - \mathbf{w}_c^*\| \leq 2G$, as in Theorem B.9, we overall have

$$\mathbb{E}f_c(\hat{\mathbf{w}}_{c,T}) - f_c(\mathbf{w}_c^*) \leq 24G^2 \frac{(s-1)s^2}{\mu T^3} + 12\sigma^2 \|\mathbf{p}_c\|^2 \frac{T+2s-1}{\mu T^2}$$

$$+ 48G^2 \|\boldsymbol{\delta}_{c,0}\|_1^2 \frac{T^{2-\beta} + s^{2-\beta}}{\mu T^3} + 48LH^2G^2 \frac{1}{\mu^2 T^2},$$

which completes the proof after rearranging the terms. $\qquad\square$

*Remark* B.11. Given uniform router initialization, we have $\|\boldsymbol{\delta}_{c,0}\|_1 = \sum_{k=1}^{K} |p(k) - \mathbf{p}_c^k| = \sum_{k=1}^{K} \mathbf{p}_c^k (1 - p(k)/\mathbf{p}_c^k) \leq (1 - U_c)$.

# C. Extending the analysis to (FML) with Weight Sharing

In this section, we show the benefits of weight sharing in the (FML) case. We now consider iterates that track the full expectation $\mathbb{E}_{k,c}$ instead of $\mathbb{E}_{k|c}$.

$$\tilde{\mathbf{w}}_t := \hat{\mathbb{E}}_{k,c}[\mathbf{w}_{c,t}^k], \qquad \tilde{\mathbf{g}}_t := \hat{\mathbb{E}}_{k,c}[\nabla f^{i_t}(\mathbf{w}_{c,t}^k)], \tag{24}$$

$$\bar{\mathbf{w}}_t := \mathbb{E}_{k,c}[\mathbf{w}_{c,t}^k], \qquad \bar{\mathbf{g}}_t := \mathbb{E}_{k,c}[\nabla f_c(\mathbf{w}_{c,t}^k)], \tag{25}$$

Note that we have assumed that $\mathbb{E}_{i_t|c}\nabla f^{i_t}(\mathbf{w}_{c,t}^k) = \nabla f_c(\mathbf{w}_{c,t}^k)$. However, we make an important distinction here. In the previous analysis in Appendix B, we have written the expectation $\mathbb{E}_{i_t|c}$, but, in fact, this $c$ is not the same as the $c$ in $\mathbb{E}_{k,c}$. The expectations $\mathbb{E}_{k,c}$ and $\hat{\mathbb{E}}_{k,c}$ track the aggregated iterates, whereas $\mathbb{E}_{i_t,c}$ takes expectation with respect to client sampling, which is independent of the tracking variables. Thus, in order to make the distinction clear, we write the sampled cluster variable as $z$ and write the expectation with respect to sampling as $\mathbb{E}_{i_t,z}$, so that $p(i_t, z = c) = \sum_{k \in i_t} p(k)\boldsymbol{\pi}_c^k$ (recall that $p(k,c) = p(k)p(c|k) = p(k)\boldsymbol{\pi}_c^k$).

Now we introduce finer variance and heterogeneity assumptions that help us achieve even better variance reduction.

**Assumption C.1** (Bounded variance of base model and adaptors). For any $c \in [C]$ and $k \in [K]$, and given weight sharing $\mathbf{w}_c = (\mathbf{u}, \mathbf{a}_c) \in \mathbb{R}^d$, we have

$$\mathbb{E}_{i_t|z=c}\|\nabla_{\mathbf{a}_c} f^{i_t}(\mathbf{w}_c) - \nabla_{\mathbf{a}_c} f_c(\mathbf{w}_c)\|^2 \leq \sigma_c^2, \tag{26}$$

$$\mathbb{E}_{i_t,z}\|\nabla_{\mathbf{u}} f^{i_t}(\mathbf{w}_c) - \mathbb{E}_{c'}\nabla_{\mathbf{u}} f_{c'}(\mathbf{w}_c)\|^2 \leq \bar{\sigma}^2. \tag{27}$$

**Assumption C.2** (Bounded heterogeneity of base model and adaptors). For $t \geq 0$, synchronization steps $t_0 \mod H = 0$, weight sharing $\mathbf{w}_{c,t}^k = (\mathbf{u}_t^k, \mathbf{a}_{c,t}^k) \in \mathbb{R}^d$, there exist $\Delta, \zeta > 0$ such that

$$\mathbb{E}_{k,c}\|\nabla_{\mathbf{u}} f_c(\mathbf{w}_{c,t}^k) - \mathbb{E}_{c'}\nabla_{\mathbf{u}} f_{c'}(\mathbf{w}_{c,t}^k)\|^2 \leq \Delta^2, \tag{28}$$

$$\hat{\mathbb{E}}_c\|\mathbf{a}_{c,t_0}^k - \hat{\mathbb{E}}_c[\mathbf{a}_{c,t_0}^k]\|^2 - 2\hat{\mathbb{E}}_c\langle\mathbf{a}_{c,t_0}^k - \hat{\mathbb{E}}_c[\mathbf{a}_{c,t_0}^k], \hat{\mathbb{E}}_{c'}\nabla_{\mathbf{a}} f_{c'}(\mathbf{w}_{c,t_0:t}^k)\rangle \leq \zeta\hat{\mathbb{E}}_{i_t,c'}\|\nabla_{\mathbf{a}} f_{c'}(\mathbf{w}_{c,t_0:t}^k)\|^2. \tag{29}$$

These assumptions do have practical relevance, especially when the fine variance quantities are smaller than the one used in Assumption 4.4. Namely, (26), which is a straightforward adaptation of Assumption 4.4, bounds the variance of the sampled *adaptors'* gradients separately (per cluster). On the other hand, (27) bounds the variance of the *base* model's gradient from the averaged objective across clusters. We expect both of these bounds to be tighter than the variance of the *full* model's gradient per cluster separately.

As for Assumption C.2, the weight sharing structure should be justified when the condition holds for small $\Delta$. The first assumption (28) bounds the (aggregated) variance of the *base* gradient across clusters, which can be close to 0 with weight sharing and small adaptors. The second assumption says that the correlation between the adaptor's signal and the gradient signal is strong enough. In particular, it should be greater than the variance of the adaptors minus some multiple of the gradient norm (of the sum of steps, which decays due to $\eta_t$). This assumption is a technical convenience to avoid bounding the adaptors's variance directly by some fixed constant, which would introduce an undesirable fixed terms in the convergence rate and would require bounded adaptors.

These assumptions help us have a more principled approach towards the practice of weight sharing and the design of adaptors. For example, we discuss a LoRA-centering procedure in Appendix F partly inspired from (29). Overall, the above assumptions decompose the variance and heterogeneity errors in a way that makes the benefits of weight sharing manifest, which is especially true using Assumption C.2.

## C.1. Analysis

We will now show that an extension of the previous analysis in Appendix B using the aforementioned quantities and assumptions is possible and can lead to better variance reduction.

In the following lemma, we will make use of the quantity $\mathbf{w}^* := \mathbb{E}_c[\mathbf{w}_c^*] = \sum_{c=1}^C p(c)\mathbf{w}_c^*$, where $p(c)$ is the overall probability of cluster $c$, e.g., see (1). This quantity is not a real optimum, but rather an analytical tool. Indeed, by Jensen's inequality, we can write $\|\tilde{\mathbf{w}}_t - \mathbf{w}^*\|^2 \leq \|\mathbf{u}_t^k - \mathbf{u}_t^*\|^2 + \mathbb{E}_c\|\mathbf{a}_{c,t}^k - \mathbf{a}_{c,t}^*\|^2$ when $\mathbf{w}_{c,t}^k = (\mathbf{u}_t^k, \mathbf{a}_{c,t}^k)$. Thus, obtaining a upper bound on the optimality gap using terms $\|\tilde{\mathbf{w}}_t - \mathbf{w}^*\|^2$ suffices as it implies the upper bound of interest.

**Lemma C.3** (Descent bound with weight sharing). *Define* $\mathbf{p} := (p(k))_{k=1}^K$ *(indexed as* $\mathbf{p}^k$*) and* $\boldsymbol{\pi}_c = (\pi_c^k)_{k=1}^K$. *Let* $\boldsymbol{\delta}_t^k = (|\pi_c^k - \hat{\pi}_{c,t}^k|)_{c=1}^C$ *and* $\mathbf{w}^* := \mathbb{E}_c[\mathbf{w}_c^*]$. *Consider the setting and assumptions in Section 4, and let Assumption C.1 and Assumption C.2 hold with* $\zeta \geq 1$, *without loss of generality. Then,*

$$\|\tilde{\mathbf{w}}_{t+1} - \mathbf{w}^*\|^2 \leq (1 - \eta_t\mu)\|\tilde{\mathbf{w}}_t - \mathbf{w}^*\|^2 + \eta_t \sum_{k=1}^K \sum_{c=1}^C \mathbf{p}^k (4L\eta_t\pi_c^k - \hat{\pi}_{c,t}^k)[f_c(\mathbf{w}_{c,t}^k) - f_c(\mathbf{w}^*t)]$$

$$+ 4\eta_t^2 (G^2\mathbb{E}_k\|\boldsymbol{\delta}_t^k\|_1^2 + \mathbb{E}_c[\|\mathbf{p}_c\|^2 \sigma_c^2] + 2\bar{\sigma}^2 \sum_{c=1}^C \|\mathbf{p} \odot \boldsymbol{\pi}_c\|^2 + 2\Delta^2)$$

$$+ 16\zeta L\eta_t^3 H^2 G^2.$$

*Proof.* As in (11), the descent can be bounded as

$$\mathbb{E}_{i_t,z}\|\tilde{\mathbf{w}}_{t+1} - \mathbf{w}^*\|^2 = \mathbb{E}_{i_t,z}\|\tilde{\mathbf{w}}_t - \mathbf{w}^* - \eta_t\bar{\mathbf{g}}_t\|^2 + \eta_t^2\mathbb{E}_{i_t,z}\|\bar{\mathbf{g}}_{c,t} - \tilde{\mathbf{g}}_t\|^2$$
$$+ 2\eta_t\mathbb{E}_{i_t,z}\langle \tilde{\mathbf{w}}_t - \mathbf{w}^* - \eta_t\bar{\mathbf{g}}_t, \bar{\mathbf{g}}_t - \tilde{\mathbf{g}}_t \rangle.$$

From the ideal aggregation descent, we have

$$\|\tilde{\mathbf{w}}_t - \mathbf{w}^* - \eta_t\bar{\mathbf{g}}_t\|^2 = \|\tilde{\mathbf{w}}_t - \mathbf{w}^*\|^2 + \eta_t^2\|\bar{\mathbf{g}}_t\|^2 - 2\eta_t\langle \tilde{\mathbf{w}}_t - \mathbf{w}^*, \bar{\mathbf{g}}_t \rangle$$
$$\leq \|\tilde{\mathbf{w}}_t - \mathbf{w}^*\|^2 + \eta_t^2\mathbb{E}_{k,c}\|\nabla f_c(\mathbf{w}_{c,t}^k)\|^2 - 2\eta_t\langle \tilde{\mathbf{w}}_t - \mathbf{w}^*, \bar{\mathbf{g}}_t \rangle.$$

As for the correlation error, we use Young's inequality and Jensen's inequality as before

$$2\eta_t\langle \tilde{\mathbf{w}}_t - \mathbf{w}^* - \eta_t\bar{\mathbf{g}}_t, \bar{\mathbf{g}}_t - \tilde{\mathbf{g}}_t \rangle = 2\eta_t\langle \tilde{\mathbf{w}}_t - \mathbf{w}^*, \bar{\mathbf{g}}_t - \tilde{\mathbf{g}}_t \rangle - 2\eta_t^2\langle \bar{\mathbf{g}}_t, \bar{\mathbf{g}}_t - \tilde{\mathbf{g}}_t \rangle$$
$$\leq 2\eta_t\langle \tilde{\mathbf{w}}_t - \mathbf{w}^*, \bar{\mathbf{g}}_t - \tilde{\mathbf{g}}_t \rangle + \eta_t^2\mathbb{E}_{k,c}\|\nabla f_c(\mathbf{w}_{c,t}^k)\|^2 + \eta_t^2\|\bar{\mathbf{g}}_t - \tilde{\mathbf{g}}_t\|^2.$$

Adding everything together (and skipping $\mathbb{E}_{i_t,z}$ for clarity), we get

$$\|\tilde{\mathbf{w}}_{t+1} - \mathbf{w}^*\|^2 \leq \|\tilde{\mathbf{w}}_t - \mathbf{w}^*\|^2 + 2\eta_t^2 \mathbb{E}_{k,c}\|\nabla f_c(\mathbf{w}_{c,t}^k)\|^2 + 2\eta_t^2\|\tilde{\mathbf{g}}_t - \bar{\mathbf{g}}_t\|^2$$
$$- 2\eta_t \hat{\mathbb{E}}_{k,c}\langle \tilde{\mathbf{w}}_t - \mathbf{w}_{c,t}^k, \nabla f_c(\mathbf{w}_{c,t}^k)\rangle - 2\eta_t \hat{\mathbb{E}}_{k,c}\langle \mathbf{w}_{c,t}^k - \mathbf{w}^*, \nabla f_c(\mathbf{w}_{c,t}^k)\rangle$$
$$\overset{(12)+(13)}{\leq} \|\tilde{\mathbf{w}}_t - \mathbf{w}^*\|^2 + 2\eta_t^2\|\tilde{\mathbf{g}}_t - \bar{\mathbf{g}}_t\|^2 - 2\eta_t \hat{\mathbb{E}}_{k,c}\langle \tilde{\mathbf{w}}_t - \mathbf{w}_{c,t}^k, \nabla f_c(\mathbf{w}_{c,t}^k)\rangle$$
$$+ \eta_t \sum_{k=1}^K \sum_{c=1}^C \mathbf{p}^k (4L\eta_t \boldsymbol{\pi}_c^k - 2\hat{\boldsymbol{\pi}}_{c,t}^k)[f_c(\mathbf{w}_{c,t}^k) - f_c(\mathbf{w}^*)] - \eta_t \mu \hat{\mathbb{E}}_{k,c}\|\mathbf{w}_{c,t}^k - \mathbf{w}^*\|^2$$
$$\leq (1 - \eta_t\mu)\|\tilde{\mathbf{w}}_t - \mathbf{w}^*\|^2 + 2\eta_t^2\|\tilde{\mathbf{g}}_t - \bar{\mathbf{g}}_t\|^2 + 2L\eta_t \hat{\mathbb{E}}_{k,c}\|\tilde{\mathbf{w}}_t - \mathbf{w}_{c,t}^k\|^2$$
$$+ \eta_t \sum_{k=1}^K \sum_{c=1}^C \mathbf{p}^k (4L\eta_t \boldsymbol{\pi}_c^k - \hat{\boldsymbol{\pi}}_{c,t}^k)[f_c(\mathbf{w}_{c,t}^k) - f_c(\mathbf{w}^*)],$$

where the last inequality uses Jensen's inequality and Young's inequality.

The optimality gap can be bounded by $-\frac{1}{2}\mathbf{p}^k \hat{\boldsymbol{\pi}}_{c,t}^k$ as in Lemma B.7 given a learning rate with a numerator $\min\{1, \min_{k\in[K]}\{\mathbf{p}^k \hat{\boldsymbol{\pi}}_{c,t}^k / \mathbf{p}^k \boldsymbol{\pi}_c^k\}\} = \min_{k; \boldsymbol{\pi}_c^k > \hat{\boldsymbol{\pi}}_{c,t}^k}\{\hat{\boldsymbol{\pi}}_{c,t}^k / \boldsymbol{\pi}_c^k\}$ this time, which allows us to obtain a bound in terms of $\hat{\mathbb{E}}_c[f_c(\tilde{\mathbf{w}}_{c,t}) - f_c(\mathbf{w}_c^*)]$.

The term $\hat{\mathbb{E}}_{k,c}\|\tilde{\mathbf{w}}_t - \mathbf{w}_{c,t}^k\|^2$ can be bounded with Lemma B.3 by adding and subtracting $\hat{\mathbb{E}}_{k,c}[\mathbf{w}_{c,t_0}^k] = \tilde{\mathbf{w}}_{t_0}$ (recall $\hat{\mathbb{E}}_{k,c}[\mathbf{u}_{c,t_0}^k] = \mathbf{u}_{c,t_0}$ and $\hat{\mathbb{E}}_{k|c}[\mathbf{a}_{c,t_0}^k] = \mathbf{a}_{c,t_0}$), applying $\text{Var}(X) \leq \mathbb{E}[X^2]$, using Assumption C.2, and then following the proof as before

$$\mathbb{E}_{i_t,z}\hat{\mathbb{E}}_{k,c}\|\mathbf{w}_{c,t}^k - \tilde{\mathbf{w}}_t\|^2 = \mathbb{E}_{i_t,z}\hat{\mathbb{E}}_{k,c}\|\mathbf{w}_{c,t}^k - \tilde{\mathbf{w}}_{t_0} - (\tilde{\mathbf{w}}_t - \tilde{\mathbf{w}}_{t_0})\|^2$$
$$\leq \mathbb{E}_{i_t,z}\hat{\mathbb{E}}_{k,c}\|\mathbf{w}_{c,t}^k - \tilde{\mathbf{w}}_{t_0}\|^2$$
$$= \mathbb{E}_{i_t,z}\hat{\mathbb{E}}_{k,c}\|\mathbf{w}_{c,t_0}^k - \tilde{\mathbf{w}}_{t_0} - \sum_{\tau=t_0}^{t-1} \eta_\tau \nabla f^{i_\tau}(\mathbf{w}_{c,\tau}^k)\|^2$$
$$= \hat{\mathbb{E}}_c\|\mathbf{a}_{c,t_0}^k - \hat{\mathbb{E}}_c[\mathbf{a}_{c,t_0}^k]\|^2 - 2\sum_{\tau=t_0}^{t-1} \eta_\tau \hat{\mathbb{E}}_c\langle \mathbf{a}_{c,t_0}^k - \hat{\mathbb{E}}_c[\mathbf{a}_{c,t_0}^k], \hat{\mathbb{E}}_z[\nabla_\mathbf{a} f_z(\mathbf{w}_{c,\tau}^k)]\rangle$$
$$+ \hat{\mathbb{E}}_{k,c}\mathbb{E}_{i_t,z}\|\sum_{\tau=t_0}^{t-1} \eta_\tau \nabla f^{i_\tau}(\mathbf{w}_{c,\tau}^k)\|^2$$
$$\overset{(29)}{\leq} (1+\zeta)\hat{\mathbb{E}}_{k,c}\mathbb{E}_{i_t,z}\|\sum_{\tau=t_0}^{t-1} \eta_\tau \nabla f^{i_\tau}(\mathbf{w}_{c,\tau}^k)\|^2 \overset{(18)}{\leq} 4(1+\zeta)\eta_t^2 H^2 G^2$$

It remains to bound $\|\tilde{\mathbf{g}}_t - \bar{\mathbf{g}}_t\|^2$. This is where the benefits of weight sharing will mainly manifest. We start bounding $\|\tilde{\mathbf{g}}_t - \bar{\mathbf{g}}_t\|^2$ as in Lemma B.2

$$\mathbb{E}_{i_t,z}\|\bar{\mathbf{g}}_t - \tilde{\mathbf{g}}_t\|^2 \leq 2\mathbb{E}_{i_t,z}\|\mathbb{E}_{k,c}[\nabla f_c(\mathbf{w}_{c,t}^k) - \nabla f^{i_t}(\mathbf{w}_{c,t}^k)]\|^2$$
$$+ 2\mathbb{E}_{i_t,z}\|\sum_{k=1}^K \sum_{c=1}^C p(k)(\boldsymbol{\pi}_c^k - \hat{\boldsymbol{\pi}}_{c,t}^k)[\nabla f_c(\mathbf{w}_{c,t}^k) - \nabla f^{i_t}(\mathbf{w}_{c,t}^k)]\|^2$$
$$\leq 2\mathbb{E}_{i_t,z}\|\mathbb{E}_{k,c}[\nabla f_c(\mathbf{w}_{c,t}^k) - \nabla f^{i_t}(\mathbf{w}_{c,t}^k)]\|^2$$
$$+ 2\sum_{k=1}^K p(k)\mathbb{E}_{i_t,z}\|\sum_{c=1}^C (\boldsymbol{\pi}_c^k - \hat{\boldsymbol{\pi}}_{c,t}^k)[\nabla f_c(\mathbf{w}_{c,t}^k) - \nabla f^{i_t}(\mathbf{w}_{c,t}^k)]\|^2$$
$$\overset{17}{\leq} 2\mathbb{E}_{i_t,z}\|\mathbb{E}_{k,c}[\nabla f_c(\mathbf{w}_{c,t}^k) - \nabla f^{i_t}(\mathbf{w}_{c,t}^k)]\|^2 + 2G^2\mathbb{E}_k\|\boldsymbol{\delta}_t^k\|_1^2.$$

By noting that $\|(\mathbf{u}, \mathbf{a}_c)\|^2 = \|\mathbf{u}\|^2 + \|\mathbf{a}_c\|^2$, the first term can be decomposed further

$$2\mathbb{E}_{i_t,z}\|\mathbb{E}_{k,c}[\nabla f_c(\mathbf{w}_{c,t}^k) - \nabla f^{i_t}(\mathbf{w}_{c,t}^k)]\|^2 = 2\mathbb{E}_{i_t,z}\|\mathbb{E}_{k,c}[\nabla_\mathbf{u} f_c(\mathbf{w}_{c,t}^k) - \nabla_\mathbf{u} f^{i_t}(\mathbf{w}_{c,t}^k)]\|^2$$
$$+ 2\mathbb{E}_{i_t,z}\|\mathbb{E}_{k,c}[\nabla_{\mathbf{a}_c} f_c(\mathbf{w}_{c,t}^k) - \nabla_{\mathbf{a}_c} f^{i_t}(\mathbf{w}_{c,t}^k)]\|^2.$$

The adaptor's term can be bounded as follows

$$2\mathbb{E}_{i_t,z}\|\mathbb{E}_{k,c}[\nabla_{\mathbf{a}_c}f_c(\mathbf{w}_{c,t}^k) - \nabla_{\mathbf{a}_c}f^{i_t}(\mathbf{w}_{c,t}^k)]\|^2$$

$$\overset{\text{(Jensen)}}{\leq} 2\sum_{c=1}^{C}p(c)\mathbb{E}_z\mathbb{E}_{i_t|z=c}\|\sum_{k=1}^{K}\mathbf{p}_c^k\left(\nabla_{\mathbf{a}_c}f_c(\mathbf{w}_{c,t}^k) - \nabla_{\mathbf{a}_c}f^{i_t}(\mathbf{w}_{c,t}^k)\right)\|^2$$

$$\overset{(26)}{\leq} 2\sum_{c=1}^{C}p(c)\|\mathbf{p}_c\|^2\sigma_c^2,$$

For the base model's term, we decompose it further

$$2\mathbb{E}_{i_t,z}\|\mathbb{E}_{k,c}[\left(\nabla_{\mathbf{u}}f_c(\mathbf{w}_{c,t}^k) - \nabla_{\mathbf{u}}f^{i_t}(\mathbf{w}_{c,t}^k)\right)]\|^2$$

$$\leq 4\mathbb{E}_{i_t,z}\|\mathbb{E}_{k,c}[\nabla_{\mathbf{u}}f_c(\mathbf{w}_{c,t}^k) - \mathbb{E}_{c'}\nabla_{\mathbf{u}}f_{c'}(\mathbf{w}_{c,t}^k)]\|^2 \tag{*}$$

$$+ 4\mathbb{E}_{i_t,z}\|\mathbb{E}_{k,c}[\mathbb{E}_{c'}\nabla_{\mathbf{u}}f_{c'}(\mathbf{w}_{c,t}^k) - \nabla_{\mathbf{u}}f^{i_t}(\mathbf{w}_{c,t}^k)]\|^2. \tag{**}$$

Observe that we can write $\mathbb{E}_{c'}[\nabla_{\mathbf{u}}f_{c'}(\mathbf{w}_{c,t}^k)] = \mathbb{E}_{i_{t'},z'}[\nabla_{\mathbf{u}}f^{i_{t'}}(\mathbf{w}_{c,t}^k)]$, so that

$$(**) = 4\text{Var}_{i_t,z}\left(\sum_{k=1}^{K}\sum_{c=1}^{C}\mathbf{p}^k\boldsymbol{\pi}_c^k\nabla_{\mathbf{u}}f^{i_t}(\mathbf{w}_{c,t}^k)\right) \overset{(27)}{\leq} 4\bar{\sigma}^2\sum_{c=1}^{C}\|\mathbf{p}\odot\boldsymbol{\pi}_c\|^2.$$

As for $(*)$,

$$(*) \overset{\text{(Jensen)}}{\leq} 4\mathbb{E}_{k,c}\|\nabla_{\mathbf{u}}f_c(\mathbf{w}_{c,t}^k) - \mathbb{E}_{c'}\nabla_{\mathbf{u}}f_{c'}(\mathbf{w}_{c,t}^k)]\|^2 \overset{(28)}{\leq} 4\Delta^2,$$

Adding the bounds for gradient error, we get

$$\mathbb{E}_{i_t,z}\|\bar{\mathbf{g}}_t - \tilde{\mathbf{g}}_t\|^2 \leq 2G^2\mathbb{E}_k\|\boldsymbol{\delta}_t^k\|_1^2 + 2\mathbb{E}_c[\|\mathbf{p}_c\|^2\sigma_c^2] + 4\bar{\sigma}^2\sum_{c=1}^{C}\|\mathbf{p}\odot\boldsymbol{\pi}_c\|^2 + 4\Delta^2. \tag{30}$$

We complete the proof by adding everything together and setting $\zeta \geq 1$. $\qquad\square$

**Convergence rate** The convergence rate will be almost identical to the main one except for some additional terms from our new assumptions and a finer, more precise total variation distance term, which would introduce a $\log C$ term instead of a $\log KC$ using the same steps as in Lemma B.6. Note that $\zeta$ could be chosen proportionally to $\eta_t^{-1}$ and still maintain convergence. As for the optimality gap, we would get it in terms of $\hat{\mathbb{E}}_c[f_c(\tilde{\mathbf{w}}_{c,t}) - f_c(\mathbf{w}_c^*)]$, as it is not possible to move the $\hat{\mathbb{E}}_c$ inside with Jensen's inequality since it is an average of different functions and not one function. We believe this can be remedied by a careful use of perturbed iterates $\mathbb{E}_{k,c}[\hat{\mathbb{E}}_{c'}f_{c'}(\mathbf{w}_{c,t}^k)]$, but we make no claims. Finally, recall that $\|\tilde{\mathbf{w}}_t - \mathbf{w}^*\|^2 \leq \|\mathbf{u}_t^k - \mathbf{u}_t^*\|^2 + \mathbb{E}_c\|\mathbf{a}_{c,t}^k - \mathbf{a}_{c,t}^*\|^2$ when $\mathbf{w}_{c,t}^k = (\mathbf{u}_t^k, \mathbf{a}_{c,t}^k)$, so that a bound on the perturbed iterates suffices.

Overall, we believe that obtaining a convergence rate from Lemma C.3 is straightforward given the main results in Theorem B.9 and Theorem B.10 and is not interesting in itself, so we shall omit it.

## C.2. Benefits of Weight Sharing

Using the above lemma, we will show the benefits of weight-sharing on some idealized examples with well-balanced client datasets and cluster sizes. First, recall the gradient aggregation error in (30)

$$\mathbb{E}_{i_t,z}\|\bar{\mathbf{g}}_t - \tilde{\mathbf{g}}_t\|^2 \leq 2G^2\mathbb{E}_k\|\boldsymbol{\delta}_t^k\|_1^2 + 2\mathbb{E}_c[\|\mathbf{p}_c\|^2\sigma_c^2] + 4\bar{\sigma}^2\sum_{c=1}^{C}\|\mathbf{p}\odot\boldsymbol{\pi}_c\|^2 + 4\Delta^2.$$

This descent bound follows from using the perturbed iterates in (24) and (25) and using Assumption C.1 and Assumption C.2. We now present examples based on (FL) and (CFL).

*Remark* C.4. Consider a balanced FL problem with $C = 1$ and $N^k = N/K$. Clearly, $\boldsymbol{\pi}_c^k = 1$ for all $k \in [K]$, so we trivially have $\boldsymbol{\delta}_t^k = \mathbf{0}$. Furthermore, $\mathbf{p}_c = 1/K$, so $\|\mathbf{p}_c\|^2 = \frac{1}{K}$, and $\sum_{c=1}^{C}\|\mathbf{p} \odot \boldsymbol{\pi}_c\|^2 = 1/K$. Finally, $\Delta^2 = 0$. Thus,

$$\mathbb{E}_{i_t,z}\|\bar{\mathbf{g}}_t - \tilde{\mathbf{g}}_t\|^2 \leq \frac{4\bar{\sigma}^2 + 2\sigma_1^2}{K},$$

which is the original variance reduction. Considering $C$ (independent) copies with $\boldsymbol{\pi}_c^k = 1/C$ and a uniform router initialization $\hat{\boldsymbol{\pi}}_{c,0}^k = 1/C$, and assuming that the variances of the adaptors are similar, i.e., $\sigma_1^2 = \cdots = \sigma_C^2$, we get

$$\mathbb{E}_{i_t,z}\|\bar{\mathbf{g}}_t - \tilde{\mathbf{g}}_t\|^2 \leq \frac{4\bar{\sigma}^2}{KC} + \frac{2\sigma_1^2}{K}, \tag{31}$$

where we can see the benefits of reducing the base model's variance by averaging further across $C$ copies of (FL) problems with independent sampling. Indeed, if $\sigma_c^2 = \sigma_1^2$ for all $c \in [C]$, then we would have $\sigma_1^2/K$, but the full $1/KC$ factor remains for the base model.

*Remark* C.5. Consider a balanced clustered problem with $N^k = N/K$ and $\boldsymbol{\pi}_c^k = \mathbf{1}\{k \in c\}$, so that $p(k) = \frac{1}{K}$ and $p(c) = \frac{K_c}{K}$, where $K_c = \sum_{k=1}^{K}\mathbf{1}\{k \in c\}$. Then, we have $\mathbf{p}_c^k = p(c|k)p(k)/p(c) = \frac{\mathbf{1}\{k \in c\}}{K_c}$, so $\|\mathbf{p}_c\|^2 = \frac{1}{K_c}$. Similarly, $\mathbb{E}_c[\|\mathbf{p}_c\|^2\sigma_c^2] = \frac{\sum_{c=1}^{C}\sigma_c^2}{K}$. Furthermore, we have $\sum_{c=1}^{C}\|\mathbf{p} \odot \mathbf{p}_c\|^2 = \sum_{c=1}^{C}\sum_{k=1}^{K}\frac{\mathbf{1}\{k \in c\}}{K^2} = \sum_{c=1}^{C}\frac{K_c}{K^2} = \frac{1}{K}$. Thus, when $\sigma_1^2 = \cdots = \sigma_C^2$, we have

$$\mathbb{E}_{i_t,z}\|\bar{\mathbf{g}}_t - \tilde{\mathbf{g}}_t\|^2 \leq \frac{4\bar{\sigma}^2}{K} + \frac{2\sigma_1^2}{K/C} + 2G^2\mathbb{E}_k\|\boldsymbol{\delta}_t^k\|_1^2 + 4\Delta^2. \tag{32}$$

Now consider a uniform router initialization $\boldsymbol{\pi}_c^k = 1/C$. Note $\boldsymbol{\delta}_0^k = (|\mathbf{1}\{k \in c\} - 1/C|)_{c=1}^{C}$, so $\|\boldsymbol{\delta}_0^k\|_1^2 = (\frac{C-1}{C} + (C-1)\frac{1}{C})^2 = 4\frac{(C-1)^2}{C^2}$. As for $\Delta^2$, we can only assume that it is close to 0. Otherwise, the use of weight sharing will not be motivated.

We can see from the clustered example that understanding the trade-off between the reduction in variances (via weight sharing) and the increase of $\Delta$ heterogeneity is important and allows for more principled mechanisms of weight sharing, which would be an interesting direction to explore.

# D. Router Update

## D.1. Derivation of router update for (MFL)

The update in (4) looks different from the one we use in practice. Indeed, consider the $\boldsymbol{\pi}^k$ that minimizes (MFL) for each $k \in [K]$, i.e., $\boldsymbol{\pi}^k = \arg\min_{\boldsymbol{\pi}} \sum_{c=1}^{C} \boldsymbol{\pi}_c f^k(\mathbf{w}_{c,t}^k)$. This is trivially $\boldsymbol{\pi}^k = \boldsymbol{e}_{\bar{c}}$ where $\boldsymbol{e}_i$ is basis vector of the $i$-th coordinate and $\bar{c} = \arg\min_{k \in [K]} f^k(\mathbf{w}_{c,t}^k)$, i.e., $\boldsymbol{\pi}^k$ is one-hot at the lowest loss. Thus, $\boldsymbol{\pi}^k$ will always lie at one of the vertices of $\Delta^{C-1}$.

However, consider now (MFL) with negative entropy regularization for the routers $\Gamma(\boldsymbol{\pi}^k) = \sum_{c=1}^{C} \boldsymbol{\pi}_c^k \log \boldsymbol{\pi}_c^k$. We have

$$\begin{aligned}
\boldsymbol{\pi}_t^k &= \arg\min_{\boldsymbol{\pi} \in \Delta^{C-1}} \sum_{c=1}^{C} \boldsymbol{\pi}_c f^k(\mathbf{w}_{c,t}^k) + \lambda_{\text{ent}} \sum_{c=1}^{C} \boldsymbol{\pi}_c \log \boldsymbol{\pi}_c \\
&= \arg\min_{\boldsymbol{\pi}_c \geq 0} \sum_{c=1}^{C} \boldsymbol{\pi}_c f^k(\mathbf{w}_{c,t}^k) + \lambda_{\text{ent}} \sum_{c=1}^{C} \boldsymbol{\pi}_c \log \boldsymbol{\pi}_c + \lambda_{\text{sim}}(\sum_{c=1}^{C} \boldsymbol{\pi}_c - 1) \\
&\implies \lambda_{\text{ent}} \log \boldsymbol{\pi}_{c,t}^k = -f^k(\mathbf{w}_{c,t}^k) - \lambda_{\text{ent}}C - \lambda_{\text{sim}}C.
\end{aligned}$$

Let $\lambda = \frac{\lambda_{\text{sim}}}{\lambda_{\text{ent}}}$. We either have $\lambda_{\text{sim}} = 0$ or $\sum_{c=1}^{C} \boldsymbol{\pi}_{c,t}^{k} = 1$, so

$$1 = \sum_{c=1}^{C} \boldsymbol{\pi}_{c,t}^{k} = \sum_{c=1}^{C} \exp(-\lambda_{\text{ent}}^{-1} f^k(\mathbf{w}_{c,t}^{k}) - C - \lambda C)$$

$$\exp(\lambda C) = \sum_{c=1}^{C} \exp(-\lambda_{\text{ent}}^{-1} f^k(\mathbf{w}_{c,t}^{k}) - C)$$

$$\lambda = \frac{1}{C} \log \sum_{c=1}^{C} \exp(-\lambda_{\text{ent}}^{-1} f^k(\mathbf{w}_{c,t}^{k}) - C) \implies \boldsymbol{\pi}_{c,t}^{k} = \frac{\exp(-\lambda_{\text{ent}}^{-1} f^k(\mathbf{w}_{c,t}^{k}))}{\sum_{c=1}^{C} \exp(-\lambda_{\text{ent}}^{-1} f^k(\mathbf{w}_{c,t}^{k}))}.$$

The above implies that the update (4) is, indeed, solving the following subproblem

$$\underset{\boldsymbol{\pi} \in \Delta^{C-1}}{\arg\min} \quad \sum_{c=1}^{C} \boldsymbol{\pi}_c f^k(\mathbf{w}_{c,t}^{k}) + \frac{1}{\eta_t} \sum_{c=1}^{C} \boldsymbol{\pi}_c \log \boldsymbol{\pi}_c. \tag{33}$$

## D.2. Connection to gradient descent on a Softmax-parameterized router

Here we show that using the router parameterization $\hat{\boldsymbol{\pi}}_c \propto \exp\theta_c$ and the update in Algorithm 1 produces similar updates to (4) up to second-order terms in the exponent given a uniform router. We first note that $\hat{\boldsymbol{\pi}}_c$ is invariant to constant shifts in $\theta_c$ under the parameterization given above. This equivalently means that $\hat{\boldsymbol{\pi}}_c$ is invariant to constant multiplications (as it is always normalized). Note that we do not make use of the time index $t$ as we will be concerned with a single update across cluster indices $c$, and since $k$ is arbitrary, we drop it for clarity.

First, we rederive the Jacobian of Softmax, i.e., $\frac{\partial \hat{\boldsymbol{\pi}}_{c'}}{\partial \theta_c}$ where $\hat{\boldsymbol{\pi}}_c = \frac{\exp\theta_c}{\sum_{c=1}^{C} \exp\theta_c}$. Using the fact that the gradient of LogSumExp is Softmax, i.e., $\frac{\partial}{\partial \theta_c} \log \sum_{c=1}^{C} \exp\theta_c = \hat{\boldsymbol{\pi}}_c$, we get

$$\frac{\partial \hat{\boldsymbol{\pi}}_{c'}}{\partial \theta_c} = \frac{\partial \log \hat{\boldsymbol{\pi}}_{c'}}{\partial \theta_c} \hat{\boldsymbol{\pi}}_{c'} = (\delta_{cc'} - \hat{\boldsymbol{\pi}}_c) \hat{\boldsymbol{\pi}}_{c'},$$

where $\delta_{cc'}$ equals 1 if $c = c'$, 0 otherwise.

Let $\hat{\mathbf{w}} := \sum_{c=1}^{C} \hat{\boldsymbol{\pi}}_c \mathbf{w}_c$. The gradient of (FML) with respect to $\theta_c$ is

$$\frac{\partial}{\partial \theta_c} f(\hat{\mathbf{w}}) = \sum_{c'=1}^{C} \langle \nabla f(\hat{\mathbf{w}}), \mathbf{w}_{c'} \rangle \frac{\partial \hat{\boldsymbol{\pi}}_{c'}}{\partial \theta_c}$$

$$= \sum_{c'=1}^{C} \langle \nabla f(\hat{\mathbf{w}}), \mathbf{w}_{c'} \rangle (\delta_{cc'} - \hat{\boldsymbol{\pi}}_c) \hat{\boldsymbol{\pi}}_{c'}$$

$$= \langle \nabla f(\hat{\mathbf{w}}), \sum_{c'=1}^{C} \delta_{cc'} \hat{\boldsymbol{\pi}}_{c'} \mathbf{w}_{c'} \rangle - \langle \nabla f(\hat{\mathbf{w}}), \sum_{c'=1}^{C} \hat{\boldsymbol{\pi}}_c \hat{\boldsymbol{\pi}}_{c'} \mathbf{w}_{c'} \rangle$$

$$= \hat{\boldsymbol{\pi}}_c \langle \nabla f(\hat{\mathbf{w}}), \mathbf{w}_c - \hat{\mathbf{w}} \rangle.$$

Using Taylor series expansion, we get

$$\frac{\partial}{\partial \theta_c} f(\hat{\mathbf{w}}) = \hat{\boldsymbol{\pi}}_c \langle \nabla f(\hat{\mathbf{w}}), \mathbf{w}_c - \hat{\mathbf{w}} \rangle = \hat{\boldsymbol{\pi}}_c (f(\mathbf{w}_c) - f(\hat{\mathbf{w}})) - \hat{\boldsymbol{\pi}}_c \Omega(\|\mathbf{w}_c - \hat{\mathbf{w}}\|^2).$$

If we assume low curvature and $\hat{\boldsymbol{\pi}}_c^{-1} = \Omega(\|\mathbf{w}_c - \hat{\mathbf{w}}\|^2)$ for $\hat{\boldsymbol{\pi}}_c > 0$, then the approximation becomes exact up to $\Theta(1)$. In other words, as the difference between cluster $c$ and the mixture increases, i.e., $\|\mathbf{w}_c - \hat{\mathbf{w}}\|^2$ becomes larger, we need $\hat{\boldsymbol{\pi}}_c$ to decrease at least as quickly so that it balances the second-order term out.

Let us simply assume that $\langle \nabla f(\hat{\mathbf{w}}), \mathbf{w}_c - \hat{\mathbf{w}} \rangle \approx f(\mathbf{w}_c) - f(\hat{\mathbf{w}})$ and that we reset $\theta_c$ before every update so that $\hat{\boldsymbol{\pi}}_c = 1/C$. Recall that $\theta_c$ is invariant to constant shifts. Thus, the step above will be

$$\theta_c - \eta \frac{\partial}{\partial \theta_c} f(\hat{\mathbf{w}}) \approx \theta_c - \eta \hat{\boldsymbol{\pi}}_c (f(\mathbf{w}_c) - f(\hat{\mathbf{w}})) = -\frac{\eta}{C} f(\mathbf{w}_c) + \underbrace{\frac{\eta}{C} f(\hat{\mathbf{w}}) + \frac{1}{C}}_{\text{do not depend on } c}. \tag{34}$$

This implies that $\hat{\boldsymbol{\pi}}_c \propto \exp(-\frac{\eta}{C}f(\mathbf{w}_c))$ since $\theta_c$ is shift-invariant, which is equal to (4) with the learning rate multiplied by $C$. In fact, we can remove router resetting, but it will then be related to the momentum-like router update $\hat{\boldsymbol{\pi}}_{c,t+1} \propto \hat{\boldsymbol{\pi}}_{c,t}\exp(-\eta f(\mathbf{w}_{c,t}))$ for a properly scaled $\eta$ with respect to $\sum_{\tau=0}^{t}\hat{\boldsymbol{\pi}}_{c,\tau}$. We leave this exposition for another work.

It should be noted that ignoring the second-order terms is not trivial. Nonetheless, it allowed us to make a direct connection between the updates we use in practice to the theory. We also understand now that the Softmax-parameterization is inferior when the curvature is high or when the mixed weights are far from the "active" clusters ($\mathbf{w}_c$ with large $\hat{\boldsymbol{\pi}}_c$). The second case happens, for example, when $\hat{\mathbf{w}}$ is in the origin and $\mathbf{w}_1$ and $\mathbf{w}_2$ are far and opposite to each other with $\hat{\boldsymbol{\pi}}_1 = \hat{\boldsymbol{\pi}}_2$, but this ambiguity is inherent. Thus, we conclude that the main difficulty that could face a Softmax-parameterized router trained with gradient descent is high curvature, which is a sound conclusion since the log-weights are linear approximations of the function objectives.

## E. Preconditioning LoRAs

Consider the gradient of a linear adaptive layer $\mathbf{W}+\mathbf{L} = \mathbf{W}+\mathbf{U}\mathbf{V}^\top$. Let $\mathbf{G}_\mathbf{U} := \nabla_\mathbf{U} f(\mathbf{W}+\mathbf{U}\mathbf{V}^\top)$ be the gradient w.r.t. parameter $\mathbf{U}$, and similarly for $\mathbf{V}$ and $\mathbf{W}$. Note that $\mathbf{G}_\mathbf{W} = \mathbf{G} := \nabla f(\mathbf{W}+\mathbf{U}\mathbf{V}^\top)$ because $\partial(\mathbf{W}+\mathbf{U}\mathbf{V}^\top)/\partial\mathbf{W} = \mathbf{I}$.

$$\begin{aligned}
\mathbf{U}\mathbf{V}^\top &\leftarrow (\mathbf{U}-\eta_t\mathbf{G}_\mathbf{U})(\mathbf{V}-\eta_t\mathbf{G}_\mathbf{V})^\top \\
&= \mathbf{U}\mathbf{V}^\top - \eta_t(\mathbf{U}\mathbf{G}_\mathbf{V}^\top + \mathbf{G}_\mathbf{U}\mathbf{V}^\top) + \mathcal{O}(\eta_t^2) \\
&= \mathbf{U}\mathbf{V}^\top - \eta_t(\mathbf{U}\mathbf{U}^\top\mathbf{G} + \mathbf{G}\mathbf{V}\mathbf{V}^\top) + \mathcal{O}(\eta_t^2),
\end{aligned}$$

where we used the chain rule, $\mathbf{G}_\mathbf{U} = \mathbf{G}\mathbf{V}$ and $\mathbf{G}_\mathbf{V} = \mathbf{G}\mathbf{U}$. For linear layers, we consider a specific preconditioner designed for low-rank estimation [46].

$$\mathbf{G}_\mathbf{U} \leftarrow \mathbf{G}_\mathbf{U}(\mathbf{V}^\top\mathbf{V}+\epsilon\mathbf{I})^{-1}, \qquad \mathbf{G}_\mathbf{V} \leftarrow \mathbf{G}_\mathbf{V}(\mathbf{U}^\top\mathbf{U}+\epsilon\mathbf{I})^{-1}, \tag{35}$$

for some small $\epsilon > 0$. We note that this idea has also been recently explored in the context of LoRAs [47]. The problem of learning a mixture of LoRAs can be ill-conditioned since they can learn at different rates, so we normalize their gradients to help them learn at the same rate [48]. Note that, as $\epsilon \to 0$, the scale of the dynamics of $\mathbf{U}\mathbf{V}^\top$ follows that of $\mathbf{W}$, i.e., $\mathbf{U}\mathbf{V}^\top - \eta_t(\mathbf{P}_\mathbf{U}\mathbf{G} + \mathbf{G}\mathbf{P}_\mathbf{V})$, where $\mathbf{P}_\mathbf{U} := \mathbf{U}(\mathbf{U}^\top\mathbf{U})^{-1}\mathbf{U}^\top$ is the projection matrix onto the column space of $\mathbf{U}$, and similarly for $\mathbf{V}$. For convolution layers, we scale by the Frobenius norm of the preconditioner instead, as the problem would otherwise involve finding the deconvolution of the preconditioner, which is out of the scope of this work. Since $\mathbf{U}^\top\mathbf{U}$ and $\mathbf{U}\mathbf{U}^\top$ have the same eigenvalues and thus the same norms, the change in $\mathbf{U}\mathbf{V}^\top$ will be proportional to $\frac{\mathbf{U}\mathbf{U}^\top}{\|\mathbf{U}\mathbf{U}^\top\|_F}\mathbf{G} + \mathbf{G}\frac{\mathbf{V}\mathbf{V}^\top}{\|\mathbf{V}\mathbf{V}^\top\|_F}$.

## F. Centering LoRAs

The condition (29) in Assumption C.2 is intuitive as a practical implementation detail. Indeed, Suppose that we have $C = 2$, and at synchronization, we have $\mathbf{u} = 5$, $\mathbf{a}_1 = 4$, and $\mathbf{a}_2 = 6$. If the model is $\mathbf{u}+\sum_c \boldsymbol{\pi}_c\mathbf{a}_c$, then an equivalent parameterization is $\mathbf{u} = 10$, $\mathbf{a}_1 = -1$, and $\mathbf{a}_2 = 1$, which has less variation across $\mathbf{a}_1$ and $\mathbf{a}_2$. What we have done is simply the following

$$\begin{aligned}
\mathbf{u} &\leftarrow \mathbf{u} + \mathbb{E}_c[\mathbf{a}_c], \\
\mathbf{a}_c &\leftarrow \mathbf{a}_c - \mathbb{E}_c[\mathbf{a}_c], \forall c \in [C],
\end{aligned}$$

where $p(c) = 1/2$. Since we have additive personalization, it is always possible to add and subtract arbitrary constants that will still yield the same parameterizations. Choosing $\mathbb{E}_c[\mathbf{a}_c]$ would simply center the adaptors around zero.

In case of LoRAs, this is not exactly as straightforward as it might seem. Consider a LoRA $\mathbf{a}_c = (\mathbf{U}_c, \mathbf{V}_c)$, for example. The update $\mathbf{a}_c = (\mathbf{U}_c - \mathbb{E}_c[\mathbf{U}_c], \mathbf{V}_c - \mathbb{E}_c[\mathbf{V}_c])$ would not really preserve the parameterization. We should, in fact, have that $\mathbf{U}_c\mathbf{V}_c^\top \leftarrow \mathbf{U}_c\mathbf{V}_c^\top - \mathbb{E}_c[\mathbf{U}_c\mathbf{V}_c^\top]$. It remains to get the

values of $\mathbf{U}_c$ and $\mathbf{V}_c$ individually after the reparameterization. We can take the closest such values by minimizing the quantity

$$\underset{\mathbf{U},\mathbf{V}}{\arg\min} \quad \|\mathbf{U}\mathbf{V}^\top - (\mathbf{U}_c\mathbf{V}_c^\top - \mathbb{E}_c[\mathbf{U}_c\mathbf{V}_c^\top])\|^2$$

But the solution is straightforward, as it is precisely the truncated SVD of $\mathbf{U}_c\mathbf{V}_c^\top - \mathbb{E}_c[\mathbf{U}_c\mathbf{V}_c^\top]$ (which is not unique). Namely,

$$\mathbf{U},\mathbf{\Sigma},\mathbf{V}^\top \leftarrow \text{Trunc-SVD}_r(\mathbf{U}_c\mathbf{V}_c^\top - \mathbb{E}_c[\mathbf{U}_c\mathbf{V}_c^\top]), \quad \mathbf{U}_c \leftarrow \mathbf{U}\mathbf{\Sigma}^{1/p}, \quad \mathbf{V}_c \leftarrow \mathbf{V}\mathbf{\Sigma}^{1/q}, \qquad (36)$$

where $r$ is the original rank of $\mathbf{U}_c$ and $\mathbf{V}_c$, and $p$ and $q$ are chosen such that $1/p + 1/q = 1$. The choice $p = 2$ and $q = 2$ is standard, but it is not exactly clear how to optimally choose $p$ and $q$ in case of LoRAs or in training FLoRAL models.

# G. Adaptors

## G.1. Convolution layer

Here, we explain some of the implementations of ConvLoRAs. In our experiments, we choose the channel+filter ConvLoRA, also called Balanced 2D, because it is the most parameter-efficient and have the best performance as per Table 3.

**Channel-wise** We define $\mathbf{U} \in \mathbb{R}^{c_{out} \times r \times k_1 \times k_2}$ and $\mathbf{V} \in \mathbb{R}^{r \times c_{in} \times 1 \times 1}$. Let us assume that $c_{out} \leq c_{in}$, without loss of generality. This could be seen as a linear transformation $\mathbf{U}$ of the $c_{in}$ filters to $r$ filters, followed by the a convolution layer $\mathbf{V}$ that is similar to the original one, except that it operates on $r$ filters instead. The order of the linear transformation and convolution can also be reversed adaptively so that the number of parameters is minimized. In general, the given construction is more economical in terms of added parameters when $c_{out} \leq c_{in}$. This operation can be written as

$$\mathbf{L}_{ijab}^{\text{channel}} := \sum_{k=1}^{r} \mathbf{U}_{ikab}\mathbf{V}_{kj11}, \qquad (37)$$

and the number of its parameters is $(c_{out}k_1k_2 + c_{in})r$.

**Filter-wise** The filter size of the convolution layer $(k_1, k_2)$ can be reduced to rank-1 filters by two consecutive convolutions with filter sizes $(k_1, 1)$ and $(1, k_2)$. Thus, for rank-$r$ filters, we define $\mathbf{U} \in \mathbb{R}^{c_{out} \times rc_{out} \times 1 \times k_2}$ and $\mathbf{V} \in \mathbb{R}^{rc_{out} \times c_{in} \times k_1 \times 1}$ as if we are decomposing the filter as a sum of rank-1 matrices. Thus, with some abuse of notation, we get the following low-rank layer

$$\mathbf{L}_{ijab}^{\text{filter}} := \sum_{k=1}^{r} \mathbf{U}_{i(rj+k)1b}\mathbf{V}_{(rj+k)ja1}. \qquad (38)$$

It is understood here that the evaluation of what is between the parenthesis gives the index of a single dimension. This adaptor has $(c_{out}k_2 + c_{in}k_1)c_{out}r$ parameters, which is significantly more than the channel-wise LoRA.

**Channel+filter-wise** : In case we want to combine channel-wise and filter-wise low-rank adaptation for channel-wise low rank $r_c$ and filter-wise low rank $r_f$, we define $\mathbf{U} \in \mathbb{R}^{c_{out} \times r_f r_c \times 1 \times k_2}$ and $\mathbf{V} \in \mathbb{R}^{r_f r_c \times c_{in} \times k_1 \times 1}$, and the adaptive layer becomes

$$\mathbf{L}_{ijab}^{\text{mix}} := \sum_{k_c=1}^{r_c} \sum_{k_f=0}^{r_f-1} \mathbf{U}_{i(r_f k_c + k_f)1b}\mathbf{B}_{(r_f k_c + k_f)ja1} = \sum_{k=1}^{r_f r_c} \mathbf{U}_{ik1b}\mathbf{V}_{kja1}. \qquad (39)$$

Letting $r := r_f r_c$, this formulation has $(c_{out}k_2 + c_{in}k_1)r$ parameters, which is an order of $c_{out}$ less parameters. In general, we always set $r_f = 1$ as filters are usually small. It is sufficient to beat the channel-wise implementation as can will be seen in Section 3.2.

**Reshaped linear** : We can use a regular linear LoRA by stacking the filter dimension of the convolution layer on the input or output channels, adding the LoRA, and then reshaping the layer back into the original shape. In other words, we have $\mathbf{U} \in \mathbb{R}^{c_{out}k_1k_2 \times r}$ and $\mathbf{V} \in \mathbb{R}^{r \times c_{in}}$, and the convolution LoRA would be

$$\mathbf{L}^{\text{conv}}_{ijab} := \mathbf{L}^{\text{linear}}_{(k_1 k_2 i + k_2 a + b)j}. \tag{40}$$

This layer has $(c_{out}k_1k_2 + c_{in})r$, exactly like the channel-wise LoRA.

In our implementation, we choose the channel+filter option as it is the most parameter-efficient. Indeed, let $c_{max} := \max(c_{in}, c_{out})$ and $c_{min} := \min(c_{in}, c_{out})$, and let $k_{max}$ and $k_{min}$ be defined similarly. Note that we can always construct a channel+filter-wise ConvLoRA such that it has $(c_{min}k_{max} + c_{max}k_{min})r$ parameters. Thus, one can check that this is less than $(c_{min}k_{max}k_{min} + c_{max})r$ only when we have $c_{max}/c_{min} \leq k_{max}$, which is likely satisfied as the standard for most architectures is to have $c_{max}/c_{min} \leq 2$, and clearly $k_{max} \geq 2$.

We can constrain the number of parameters similarly to the linear layer as $(c_{min}k_{max} + c_{max}k_{min})r \leq \rho c_{min}c_{max}k_{min}k_{max}$. Indeed, if $c_{max} = c_{min}$ and $k_{max} = k_{min}$, we have $r \leq \rho c_{max}k_{max}/2$. The split of kernel sizes among the two layers can be done adaptively such that $r$ is maximized. In the experiment section, we refer to channel-wise ConvLoRAs methods where $r$ is maximized given $\rho$, and similarly for the channel+filter-wise ConvLoRAs methods where $r$ is maximized given $\rho$, which we denote as Balanced 2D ConvLoRA. We show the comparisons in Figure 11 and Table 3.

## G.2. Normalization Layers

We consider adaptors to batch normalization, instance normalization, layer normalization, and group normalization. All of these normalization layers start by normalizing a hidden vector of some layer $\mathbf{h}$ along specific dimensions to get $\hat{\mathbf{h}}$ and then take a Hadamard product along the normalized dimension as $\boldsymbol{\gamma} \odot \hat{\mathbf{h}}$ (ignoring bias). We propose a simple adaptor $\mathbf{L}_\gamma$ that has the same shape and works in exactly the same manner but is initialized to zero. The adaptive output will then be $(\boldsymbol{\gamma} + \mathbf{L}_\gamma) \odot \hat{\mathbf{h}}$, which is initially equal to the non-adaptive output.

One normalization layer that requires a more thorough treatment is batch normalization. This is because it normalizes $\mathbf{h}$ with respect to running statistics calculated from previous batches, so the adaptor would need to normalize with respect to the same running statistics if we want to maintain the same additive form of the output under the same scale.

We now show a simple reparameterization of the BatchNorA that normalizes $\mathbf{h}$ with respect to the adaptor statistics but trains its parameters with respect to the main statistics. This ensures that the gradient of the adaptor has the same scale as the original gradient. This is useful because we are interested in the federated learning case where those parameters are federated, but *the statistics are local*. Note that this is not the same as FedBN [49], where both the parameters and the statistics are local.

**Batch NorA** We will show here a batch norm adaptor that might be of interest to the readers, which is left here in the appendix as it is still in the exploratory stage. Preliminary experiments show decent improvements, as can be seen from Figure 3.

First, recall batch normalization

$$\text{BN}(x; \gamma, \beta) = \frac{x - \hat{\mu}(x)}{\sqrt{\hat{\sigma}^2(x) + \epsilon}}\gamma + \beta,$$

where $x \in \mathbb{R}^{B \times d}$ for batch size $B$ and dimension $d$, $\hat{\mu}(x) \in \mathbb{R}^d$ and $\hat{\sigma}^2(x) \in \mathbb{R}^d$ are the batch mean and variance (or statistics, for short), $\gamma \in \mathbb{R}^d$ and $\beta \in \mathbb{R}^d$ are learnable parameters, and $\epsilon$ is a small number for numerical stability. Here, it is understood that the operation is applied on $x$ batch-wise. Often, batch statistics are estimated with a running (exponential) average during training, and then fixed during evaluation.

When we are faced with multiple tasks or non-iid data distributions, batch normalization layers can actually hurt performance because the batch statistics can be inaccurate and might not necessarily

converge [50]. We would like to introduce an adaptor for batch norm layers $L_i = [\gamma_i, \beta_i]$, so an intuitive implementation would be as follows:

$$\text{BN-Adaptor}_i(x; \gamma, \beta, \gamma_i, \beta_i) = \text{BN}(x; \gamma, \beta) + \text{BN}_i(x; \gamma_i, \beta_i),$$

where both $\gamma_i$ and $\beta_i$ are initialized to 0 so that it is equivalent to the original case at initialization.

However, we want to ensure that our choice of $\gamma_i$ and $\beta_i$ is invariant to the local batch statistics. In other words, we want $\gamma_i$ to behave as a perturbation to $\gamma$, and similarly for $\beta_i$. Let us set $\epsilon = 0$. Now, observe that

$$\text{BN-Adaptor}_i(x) = \frac{x - \hat{\mu}(x)}{\sqrt{\hat{\sigma}^2(x)}}\gamma + \frac{x - \hat{\mu}_i(x)}{\sqrt{\hat{\sigma}_i^2(x)}}\gamma_i + \beta + \beta_i$$

$$= \frac{x - \hat{\mu}(x)}{\sqrt{\hat{\sigma}^2(x)}}\gamma + \frac{\sqrt{\hat{\sigma}^2(x)}}{\sqrt{\hat{\sigma}_i^2(x)}}\frac{x - \hat{\mu}_i(x)}{\sqrt{\hat{\sigma}^2(x)}}\gamma_i + \beta + \beta_i$$

$$= \frac{x - \hat{\mu}(x)}{\sqrt{\hat{\sigma}^2(x)}}\gamma + \frac{\sqrt{\hat{\sigma}^2(x)}}{\sqrt{\hat{\sigma}_i^2(x)}}\left(\frac{x - \hat{\mu}(x)}{\sqrt{\hat{\sigma}^2(x)}} - \frac{\hat{\mu}_i(x) - \hat{\mu}(x)}{\sqrt{\hat{\sigma}^2(x)}}\right)\gamma_i + \beta + \beta_i.$$

Let $\hat{m}_i := \frac{\hat{\mu}_i(x) - \hat{\mu}(x)}{\sqrt{\hat{\sigma}^2(x)}}$ be the (normalized) mean shift w.r.t. the global mean and $\hat{s}_i := \frac{\hat{\sigma}_i(x)}{\hat{\sigma}(x)}$ be the relative deviation w.r.t. the global deviation. We can rewrite the above expression as

$$\text{BN-Adaptor}_i(x) = \frac{x - \hat{\mu}(x)}{\sqrt{\hat{\sigma}^2(x)}}\gamma + \hat{s}_i^{-1}\left(\frac{x - \hat{\mu}(x)}{\sqrt{\hat{\sigma}^2(x)}} - \hat{m}_i\right)\gamma_i + \beta + \beta_i$$

$$= \frac{x - \hat{\mu}(x)}{\sqrt{\hat{\sigma}^2(x)}}(\gamma + \hat{s}_i^{-1}\gamma_i) - \hat{m}_i\hat{s}_i^{-1}\gamma_i + \beta + \beta_i.$$

Thus, consider a reparameterization $\tilde{\gamma}_i := \hat{s}_i\gamma_i$ and $\tilde{\beta}_i := \beta_i + \hat{m}_i\gamma_i$ so that $\text{LoRA-BN}_i(x) = \text{BN}(x; \gamma, \beta) + \text{BN}_i(x; \tilde{\gamma}_i, \tilde{\beta}_i)$. We would then have that

$$\text{BN-Adaptor}_i(x) = \frac{x - \hat{\mu}(x)}{\sqrt{\hat{\sigma}^2(x)}}(\gamma + \hat{s}_i^{-1}\tilde{\gamma}_i) + \hat{m}_i\hat{s}_i^{-1}\tilde{\gamma}_i + \beta + \tilde{\beta}_i$$

$$= \frac{x - \hat{\mu}(x)}{\sqrt{\hat{\sigma}^2(x)}}(\gamma + \gamma_i) + \beta + \beta_i.$$

Therefore, a reparameterization that is invariant to local batch statistics would be as follows

$$\gamma_i \longrightarrow \frac{\hat{\sigma}_i(x)}{\hat{\sigma}(x)}\gamma_i, \qquad \beta_i \longrightarrow \beta_i + \frac{\hat{\mu}_i(x) - \hat{\mu}(x)}{\sqrt{\hat{\sigma}^2(x)}}\text{sg}(\gamma_i), \tag{41}$$

where we used the stop gradient operator $\text{sg}(\gamma_i)$ to emphasize that $\gamma_i$ is given in $\beta_i$'s parameterization (i.e., would not pass its gradients through $\beta_i$). Note that this $\gamma_i$ is **not** the reparameterized one. It is helpful to think of the expressions on the RHS of the arrows in (41) as arguments to the batch norm function, and that $\gamma_i$ and $\beta_i$ are parameters to be optimized.

**Experiment** Consider the following small adjustment to the synthetic MLP task. For each client $k$, we first take a fixed sample of $\mathbf{x}^k$, compute the hidden vectors, and then normalize them before feeding them to the activation function and final layer. The normalization is critically dependent on the sampled $\mathbf{x}^k$ for each client. This construction makes the problem more amenable to a batch normalization layer after the first layer, so we use this model and consider Batch-NorA. In addition, we consider use batch normalization in the VGG-8 model we originally used for CIFAR-100.

The results in Figure 3 are decent and show that the particular setting of reparameterized Batch-NorA Appendix G.2 with local statistics can offer good improvements. We note that the reparameterization is equivalent to normalization with respect to the main batch norm and then rescaling and shifting with respect to the adaptor's parameters. The convenience of this reparameterization is that it does not require any adjustment to the batch norm layer in the adaptor, and the reparameterization can be seamlessly done with PyTorch's parameterization module.

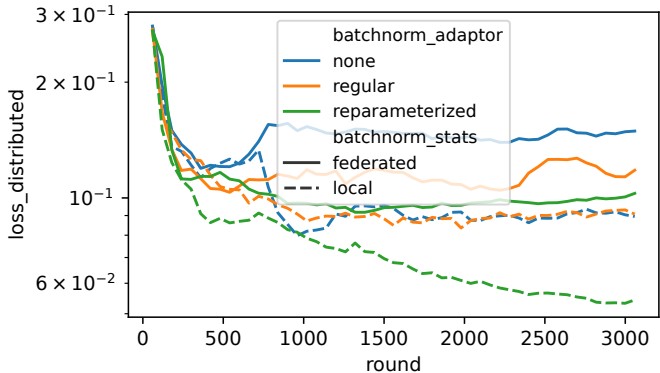

Figure 3: Loss on Synthetic MLP + BN dataset.

# H. Extra Experimental Details

In this section, we show extra experimental details and show missing tables and figures.

## H.1. Synthetic Linear

Consider a regression task where we want to learn $\mathbf{y} \in \mathbb{R}^{d_y}$ given $\mathbf{x} \in \mathbb{R}^{d_x}$, where $\mathbf{x} \sim \mathcal{N}(0, \mathbf{I}_{d_x})$. We construct two versions of this regression task: one is based on a linear model plus a personalized LoRA, and the other is based on a similar setup on the first layer of a two-layer ReLU net. For both problems, we sample the parameters of the dataset element-wise from the normal distribution $\mathcal{N}(0, \frac{1}{\sqrt{d_{in}}})$, where $d_{in}$ is the input dimension of the layer.

The target and the model are such that

$$\mathbf{y}^k(\mathbf{x}) = \sum_{c=1}^{C} \boldsymbol{\pi}_c^k(\mathbf{W} + \alpha \mathbf{U}_c \mathbf{V}_c^\top)\mathbf{x}, \qquad \hat{\mathbf{y}}^k(\mathbf{x}) = \sum_{c=1}^{C} \hat{\boldsymbol{\pi}}_c^k(\hat{\mathbf{W}}^k + \hat{\mathbf{U}}_c^k(\hat{\mathbf{V}}_c^k)^\top)\mathbf{x}, \tag{42}$$

where $\mathbf{W} \in \mathbb{R}^{d_y \times d_x}$, $\mathbf{U}_c \in \mathbb{R}^{d_y \times r}$, $\mathbf{V}_c \in \mathbb{R}^{d_x \times r}$, and $\alpha \in \mathbb{R}$, and similarly for the trained parameters. The ground-truth model is designed such that the clients share a common structure without making any assumption about the distances of the personal solutions to the solution of (FL). Notice that $\alpha$ can make the personal solutions arbitrarily far from $\mathbf{W}$, yet they differ in rank $r$ only. For example, a simple construction would be $\mathbf{W} = \mathbf{I}$ and $\mathbf{U}_c = \mathbf{V}_c = \mathbf{e}_c$, where $\mathbf{e}_i$ is the standard basis vector of the $i$-th coordinate (e.g., $\mathbf{e}_1 = (1, 0, \cdots)^\top$). As for the ground-truth router assignment, we consider a *diagonal* assignment such that $\boldsymbol{\pi}_c^k = \delta_{(k \mod C)c}$, so clients $mk$ are in the same cluster for positive integers $m$.

For each client $k$, we take a fixed sample of $\mathbf{x}^k$ and $\mathbf{y}^k$ of size $N^k$ such that $N^k < d$, but $\sum_{k=1}^{K} N^k > d$, where $d = d_y d_x$ is the original model size. This is to make it difficult for the model to fit $\hat{\mathbf{W}}$ locally due to under-parameterization. Thus, collaboration is important to generalize well, but collaboration with the wrong clients can be detrimental. For this dataset, we chose $N^k \approx 0.25d$. The objective for this regression task is the MSE loss $\frac{1}{2}\|\hat{\mathbf{y}}^k(\mathbf{x}) - \mathbf{y}^k\|^2$..

## H.2. Synthetic MLP

Consider a 2-layer ReLU neural net, or multi-layer perceptron (MLP) for short[6]

$$\mathbf{y}^k := \boldsymbol{\Phi}\left(\sum_{c=1}^{C} \boldsymbol{\pi}_c^k(\mathbf{W} + \mathbf{U}_c \mathbf{V}_c^\top)\mathbf{x}\right)_+, \tag{43}$$

---

[6]We write the ReLU function as $(\cdot)_+$.

Table 3: Ablation of ConvLoRAs.

| ConvLoRA | CIFAR-10 | | CIFAR-100 |
|---|---|---|---|
| | R | LS | |
| Balanced 2D | **70.2** | **74.1** | *51.7* |
| In Layer | 67.6 | 73.5 | 49.1 |
| Out Layer | *68.5* | *74.0* | **51.9** |
| None | 67.6 | 73.9 | 50.8 |

Table 4: Ablation of adaptors.

| Adaptors | Bias | CIFAR-10 | | CIFAR-100 |
|---|---|---|---|---|
| | | R | LS | |
| ConvLoRA | ✗ | 69.8 | 72.7 | 45.1 |
| | ✓ | 67.6 | 73.4 | 45.8 |
| LoRA | ✗ | 68.7 | 73.7 | 46.6 |
| | ✓ | 67.6 | 73.9 | *50.8* |
| Both | ✗ | *68.9* | 73.3 | 47.9 |
| | ✓ | **70.2** | **74.1** | **51.7** |
| None | ✗ | 64.6 | 21.9 | 12.1 |
| | ✓ | 64.6 | 21.9 | 12.1 |

Table 5: Metadata of the considered federated datasets ($K$ = # of clients, $C$ = # of clusters, $p$ = ratio of sampled clients per round).

| Dataset | $K$ | $C$ | $p$ | Model |
|---|---|---|---|---|
| Synthetic Linear | 10 | 2 | 100% | Linear (42) |
| Synthetic MLP | 20 | 4 | 100% | MLP (43) |
| MNIST | 300 | 4 | 10% | MLP |
| CIFAR-10 | 20 | 4 | 100% | 2×Conv→MLP |
| CIFAR-100 | 100 | 10 | 50% | VGG-8 |

where now $\mathbf{W} \in \mathbb{R}^{d_h \times d_x}$, $\mathbf{U}_c \in \mathbb{R}^{d_h \times r}$, $\mathbf{V}_c \in \mathbb{R}^{d_x \times r}$, and $\mathbf{\Phi} \in \mathbb{R}^{d_y \times d_h}$ for some hidden dimension $d_h$, and a diagonal router assignment $\pi_c^k = \delta_{(k \mod C)c}$. We use normal initialization with variance proportional to the input dimension of the layer.

The regression model has the exact same form. However, the hidden dimension is wider, i.e., it is $md_h$ for some integer $m \geq 1$. This is mainly because we want to control for the effect of not being able to fit the target model (we set $m = 2$ in our experiments). We also have $N^k \approx 0.5d$, which is twice as many data points than the linear task as this task is more difficult.

## H.3. Ablation and Hyperparameters

**Adaptors.** We study the effect of removing each of the adaptors introduced in Section 3.2. We chose the CIFAR-10 with both tasks and CIFAR-100 for the ablation study of the LoRAs, ConvLoRAs, and bias adaptors. We show in Figure 9 and Table 4 that the full combination of LoRA, ConvLoRA, and adaptive biases can consistently achieve the top accuracy.

$\rho$ **and** $C$**.** In Table 2, we see that choosing $C$ to be less than the number of ground-truth clusters can hurt performance. On the other hand, using a significantly larger $C$ can hurt performance for smaller $\rho$, but a larger $\rho$ fixes this by reaching similar accuracies to the case where we know the exact number of ground-truth clusters. We can also see the plots in Figure 10.

**ConvLoRA.** We compare the different methods for implementing ConvLoRAs as proposed in Section 3.2. We propose to balance the channels and the kernel sizes such that we achieve the most parameter-efficient ConvLoRA, which we refer to as Balanced 2D as it is specific to the two dimensional case. On the other hand, we can balance only the channels and fix the kernel sizes to either the in layer or the out layer. We show in Table 3 and Figure 11 that the Balanced 2D case is consistently the best option given a fixed $\rho$. Recall that MNIST and CIFAR-10 have 4 ground-truth clusters, and CIFAR-100 have 10.

## H.4. Datasets Meta-data

See Table 5.

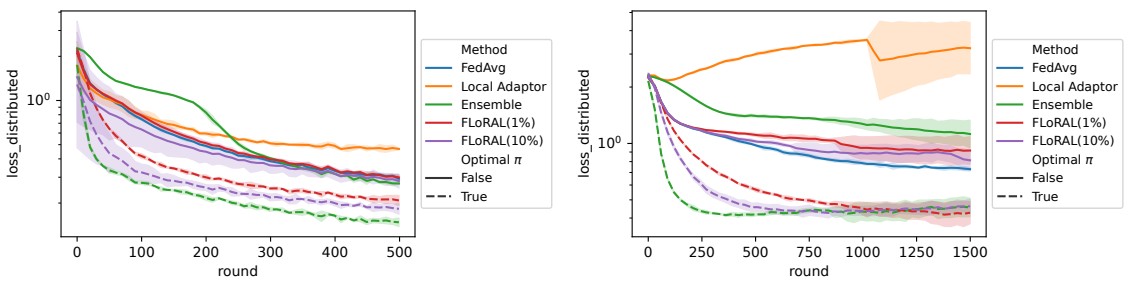

Figure 4: Test loss on MNIST-R (left = Full, right = Reduced).

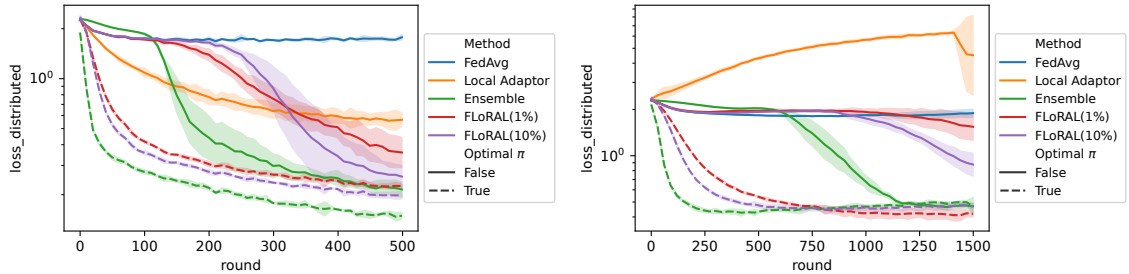

Figure 5: Test loss on MNIST-LS (left = Full, right = Reduced).

## H.5. Missing Figures

In this section, we simply show missing figures from our experiments for completeness. In particular, we show plots of the aggregated testing loss per client, which shows how the other methods overfit in comparison to FLoRAL, especially in the low-data regime.

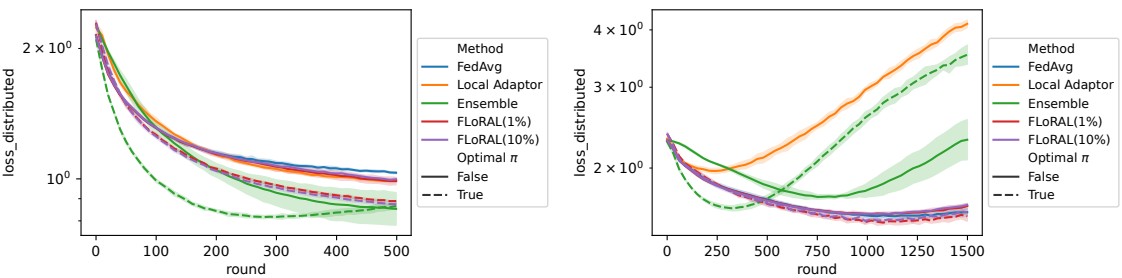

Figure 6: Test loss on CIFAR-10-R (left = Full, right = Reduced).

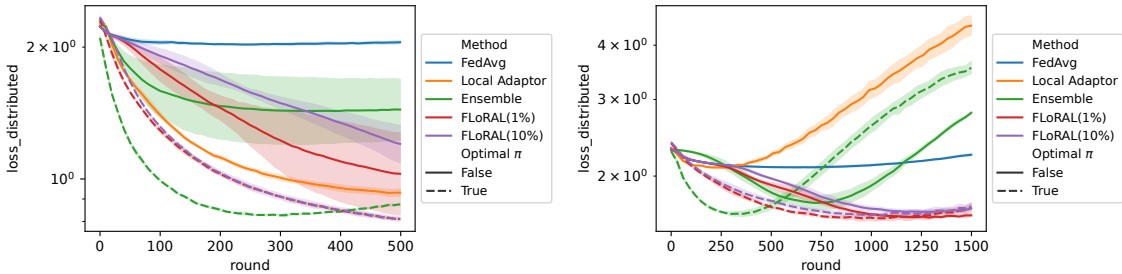

Figure 7: Test loss on CIFAR-10-LS (left = Full, right = Reduced).

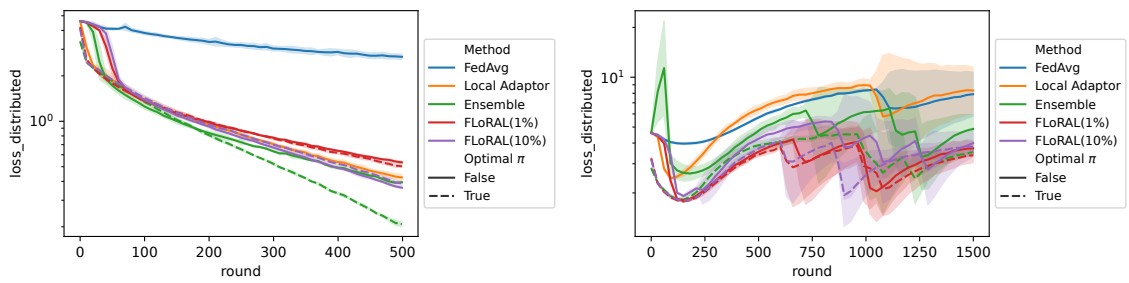

Figure 8: Test loss on CIFAR-100 (left = Full, right = Reduced).

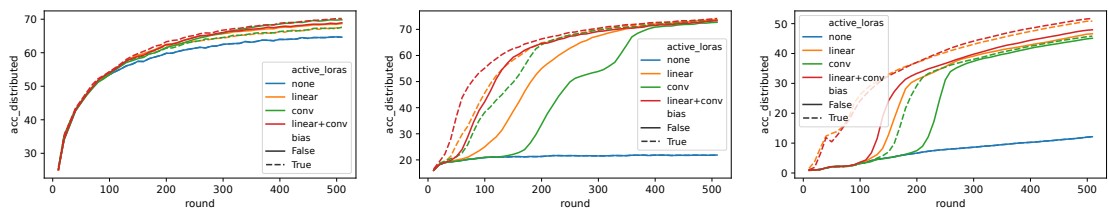

Figure 9: Ablation study of FLoRAL Adaptors (left: CIFAR-10-R, middle: CIFAR-10-LS, right: CIFAR-100).

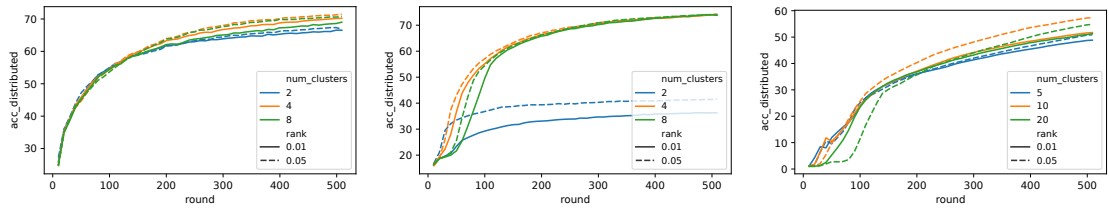

Figure 10: Varying $\rho$ and $C$ (left: CIFAR-10-R, middle: CIFAR-10-LS, right: CIFAR-100).

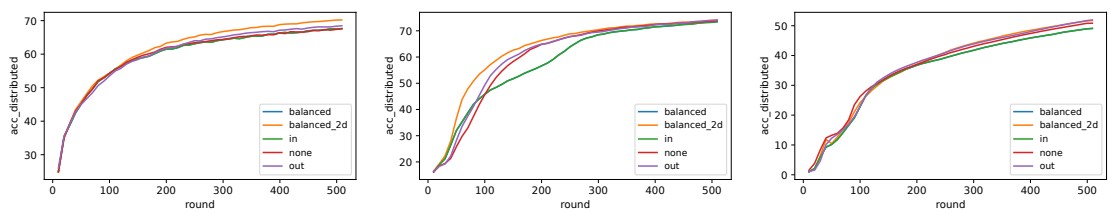

Figure 11: Accuracy of ConvLoRA as described in Appendix H.3 (left: CIFAR-10-R, middle: CIFAR-10-LS, right: CIFAR-100).

