# OpenReview forum: "Collaborative and Efficient Personalization with Mixtures of Adaptors"
_CPAL.cc/2025/Proceedings_Track — CPAL 2025 (Proceedings Track) Poster_

### Official Review · Reviewer_JTsv · 2025-01-10
**A federated learning algorithm via mixture of LoRA**

**Rating:** 5
**Confidence:** 4

**Review:**

This paper proposes to improve the efficiency of Federated Learning by learning with a mixture of low-rank adaptors. This idea is rather straight-forward. While I don't downgrade the merits of papers with simple solutions, I didn't find too many valuable insights in this paper. I encourage the authors to clarify the challenges of applying mixture of low-rank adaptors to federated learning, as similar methods have already been applied to other fields.

---

### Official Review · Reviewer_q55P · 2025-01-14
**Solid Work Addressing the Critical Challenge of Parameter-Efficient Personalization in Federated Learning**

**Rating:** 8
**Confidence:** 3

**Review:**

This paper introduces a framework named Federated Low-Rank Adaptive Learning (FLoRAL) for parameter-efficient personalized federated learning (FL). It proposes a novel approach to tackle the heterogeneity of client data in FL by leveraging low-rank adaptors and personalized model mixtures.

Strength:
- This paper focus on the interesting and important topic: parameter-efficient personalization in federated learning.
- The paper provides solid therotical convergence rate analysis and highlights variance reduction benefits.
- It includes thorough empirical evaluations and ablation studies, comparing FLoRAL with existing methods such as FedAvg, local adaptors, and ensemble models.

---

### Official Review · Reviewer_mCjd · 2025-01-15

**Rating:** 7
**Confidence:** 2

**Review:**

## Summary
The paper introduces FLoRAL, a parameter-efficient framework that balances collaboration and personalization by employing parameter-efficiet low-rank adaptors. The method models client heterogeneity through a mixture of shared cluster models, where each cluster model consists of a shared component and a personalized adaptor. The authors show analysis on convergence of their method and several practical tricks.
## Pros
1. the method reduces the number of parameters required for personalization in application of federal learning. The proposed direction is promising.
2. The weight-sharing promotes efficiency in parameter usage across clusters and encourage local clients to gain information from multiple data resources.
3. The paper presents a solid theoretical foundation, including the convergence of local SGD on a relaxed objective. The authors discuss the effects of aggregation mismatch on convergence, which adds depth to the understanding of the framework's behavior.
## Cons
1. Although the method involves the use of LoRA, the experimental results rely on additional techniques, such as preconditioning LoRA. While these techniques are intuitively beneficial, the authors do not clearly demonstrate whether they are essential. The key contribution of the paper is the proposal of FLoRAL for FL personalization, but the experimental evaluation does not directly verify the method without applying these additional tricks.
2. The experimental setup lacks clarity regarding the choice of baselines. Specifically, the rationale for selecting only FedAvg, Local Adaptor, and Ensemble as baselines is not well explained. Including additional baselines, such as [1], could strengthen the evaluation.
3. Some typographical error exists. For instance, in Equation 1 of Section 4, the notations $\pi_{k' c}$ and $N_{k'}$ could be $\pi^{k'}_c$ and $N^{k'}$?
## Quality, clarity and originality
1. The paper presents clear theoretical analysis. I think the main issue lies on its presentation of the experimental evaluation.
2. No ethics review needed.

[1] Wu, X., Liu, X., Niu, J., Wang, H., Tang, S., & Zhu, G. (2024). FedLoRA: When Personalized Federated Learning Meets Low-Rank Adaptation.

---

### Meta-Review · Area_Chair_MQcb · 2025-02-02

**Recommendation:** Accept (Poster)
**Confidence:** 3

**Metareview:**

Reviewers broadly appreciate this paper’s clear motivation and theoretical contributions for parameter-efficient personalization in federated learning. Two reviewers strongly recommend acceptance, noting the solid convergence analysis and consistent empirical improvements over relevant baselines. One reviewer expressed concern about novelty and requested more clarity regarding the use of low-rank adaptors; however, the authors’ rebuttal seems to address these points to a satisfactory degree. Overall, the consensus is that the paper provides an interesting contribution to the field and should be accepted.

---

### Decision · Program_Chairs · 2025-02-11

Accept (Poster)